

**Model-based analysis of erosion-induced microplastic delivery from arable land to the**
**stream network of a mesoscale catchment**
Raphael Rehm[a], Peter Fiener[a]
[a] University of Augsburg, Institute of Geography, *Alter Postweg 118, 86159 Augsburg,*
*Germany*
*Correspondance to*: Peter Fiener (peter.fiener@geo.uni-ausburg.de)





**Abstract**
Soils are generally accepted as sinks for microplastic (MP), but at the same time might be a MP source
for inland waters. However, little is known regarding the potential MP delivery from soils to aquatic
systems via surface runoff and erosion. This study provides for the first time an estimate of the extent of
soil erosion-induced MP delivery from an arable-dominated mesoscale catchment (390 km²) to its river
network within a typical arable region of Southern Germany. To do this, a soil erosion model was used
and combined with potential particular MP load on arable land from different sources (sewage sludge,
compost, atmospheric deposition and tyre wear) since 1950. The modelling resulted in an annual mean
MP flux into the stream network of 6.33 kg in 2020, which was dominated by tyre wear (80%). Overall,
0.11–0.17% of the MP applied to arable soils between 1950 and 2020 was transported into the stream
network. In terms of mass, this small proportion was in the same range as the MP inputs from wastewater
treatment plants within the test catchment. More MP (0.5–1% of input between 1950 and 2020) was
deposited in the grassland areas along the stream network, and this could be an additional source of MP
during flood events.  Most (5% of the MP applied between 1950 and 2020) of the MP translocated by
tillage and water erosion was buried under the plough layer. Thus, the main part of the MP added to
arable land remained in the topsoil and is available for long-term soil erosion. This can be illustrated
based on a 'stop MP input in 2020' scenario, indicating that MP delivery to the stream network until
2100 would only be reduced by 14%. Overall, arable land at risk of soil erosion represents a long-term
MP sink, but also a long-term MP source for inland waters.



## 1. Introduction

The global microplastic (MP) contamination of different environmental compartments is currently the
focus of different research fields (Nasseri and Azizi, 2022; Tian et al., 2022; Zhang et al., 2022). Among
these, MP in soils have increasingly received scientific attention (Chia et al., 2021; Sajjad et al., 2022;
Zhou et al., 2020). Many MP sources have been identified for soil systems. Next to tyre wear (TW),
stated as the main source (Knight et al., 2020; Sommer et al., 2018), littering (Scheurer and Bigalke,
2018) and atmospheric deposition (Brahney et al., 2020) also serve as MP input pathways. Arable soils
in particular experience increased MP inputs as a result of agricultural management (Brandes, 2020).
Mulch films (Ng et al., 2020), the use of compost and sewage sludge as organic fertilizers (Braun et al.,
2021; Liu et al., 2014; Zhang et al., 2020), irrigation with contaminated (waste) water (Pérez-Reverón et
al., 2022), as well as MP associated with coated fertilizer and seeds (Accinelli et al., 2021; Lian et al.,
2021), have proven to be the main input paths. MP enters the soil system mostly via the surface and is
mixed into the soil column via bioturbation (Heinze et al., 2022; Li et al., 2021) and, in the case of small
particles, via infiltration (Li et al., 2021). In arable land, it is actively mixed into the plough layer via
tillage operations (Weber et al., 2022; Zhao et al., 2022; Zubris and Richards, 2005). Depending on the
tillage technique, the MP is worked into the soil at different depths and is more or less homogenized after
multiple processing (Fiener et al., 2018; Weber et al., 2022). Moreover, tillage potentially leads to
mechanical fragmentation of macroplastic but also reduces photochemical decomposition at the soil
surface and reduces MP transport via water and wind (Colin et al., 1981; Corcoran, 2022; Feuilloley et
al., 2005).
Despite the known pathways into the soil, knowledge of the fate of MP particles once they enter the
soil system is limited (Guo et al., 2020; Hurley and Nizzetto, 2018; Tian et al., 2022). However, the
question arises as to whether the terrestrial MP sink releases relevant amounts of MP for water bodies
via water erosion. If so, the soils, as an MP sink, could represent an important MP source for water



bodies. Besides very slow, not very well determined processes of plastic fragmentation (Corcoran, 2022),
there is also only a small number of studies analysing vertical MP transport due to bioturbation (Heinze
et al., 2022; Li et al., 2021) and leaching (Chia et al., 2021; Viaroli et al., 2022) within the soil column,
or lateral losses to other ecosystems via erosion processes (Borthakur et al., 2022; Bullard et al., 2021;
Rehm et al., 2021).

The potential lateral transport via (water) erosion processes might be analysed using existing

modelling techniques. Such approaches face two major challenges: modelling approaches are required
which allow the cumulative loss of MP to adjacent ecosystems to be determined while taking spatial
differences in MP contamination and site-specific erosion into account. Moreover, the long-term change
in MP concentrations in the plough layer should be considered, following mixing with subsoil at
erosional sites or burial of MP below the plough layer at depositional sites.

In general, there are different water erosion modelling approaches available, ranging from physically-

oriented models (MCST, Fiener et al., 2008; e.g. EROSION3D, Schmidt et al., 1999), which might be
suitable for dealing with the specific particle size and density of MP during transport in the case of
individual erosion events, to conceptual approaches (e.g. WaTEM/SEDEM, (Van Oost et al., 2000; Van
Rompaey et al., 2001), which are able to consider long-term cumulative MP soil contamination and the
associated long-term soil and MP erosion, transport and deposition. In general, models of the first type
are very parameter and input data intensive and are mostly applied in small catchments, while the second
type of model needs less detailed data and is often used for mesoscale catchments (Nunes et al., 2018).
Following the requirements outlined above, conceptual, long-term approaches that account for spatial
variability in MP soil contamination and erosion processes seemed to be more appropriate than process-
oriented models to simulate the magnitude of erosion-induced MP delivery to the stream network of
mesoscale catchments. As MP loss below the plough layer might be also important in reducing topsoil
MP contamination, such a model approach should not only simulate water erosion, but also tillage





erosion processes leading to a reduction of the MP concentration at erosional sites and MP burial below
the plough layer at depositional sites. One of the few models simulating long-term water and tillage
erosion in a spatial context that updates the soil properties within the soil profile is the SPEROS-C model
(Fiener et al., 2015; Van Oost et al., 2005b). The water and tillage erosion components of the model,
originating from the WaTEM/SEDEM model (Van Oost et al., 2000; Van Rompaey et al., 2001), were
tested in several micro- and mesoscale catchments (Krasa et al., 2005; Verstraeten and Prosser, 2008).
The general objective of this study is to investigate MP transport from arable land to the stream
network in an example mesoscale (390 km²) arable catchment in Southern Germany. Therefore, the
SPEROS-C carbon transport model was adjusted to study the importance of water and tillage erosion
processes for particular MP transport. Specifically, this study focuses on the following areas: (i)
quantifying the importance of the water erosion pathway for MP input to the stream network in an
example mesoscale catchment, while taking into account the large uncertainties, particularly in estimates
of MP input to soil; (ii) determining the importance of different erosion processes in changing the MP
concentration in the plough layer and burying MP below the plough layer, and (iii) using scenarios to
determine future pathways of diffuse MP delivery into the stream network.
**2. Methods**
*2.1. Test catchment*
The catchment was chosen for two main reasons: (i) it represents an intensively used arable landscape
in Southern Germany with hilly terrain and highly productive, loess-burden soils, and (ii) the Bavarian
States Office for Environment has monitored discharge and sediment delivery at the outlet since 1968,
which allows the erosion component of the model to be tested. The mesoscale Glonn catchment
(48°22'N, 11°24'E) covers 390 km² and its altitude ranges from 578 m in its south-west to 447 m a.s.l.
at its outlet in the north-east (Fig. 1). Mean annual temperature and mean precipitation of the region are





7.5°C and 876 mm respectively, with the most intense rainfall events associated with convective rainfall
in summer. The hilly landscape (4.7±3.7° main slope) is characterized by loamy Cambisols (WRB, 2015)
on the elevated terrain and loamy Gleysols (WRB, 2015) in the valleys. Land cover in this area is
dominated by arable land (54%), followed by forest (21%), grassland (14%) and settlements (11%) (Fig.
1). The main crops are arranged in a corn-grain rotation. Due to the topography and the soils, erosion
rates reach values of about 10 t ha$^{-1}$ a$^{-1}$ (Auerswald et al., 2009).

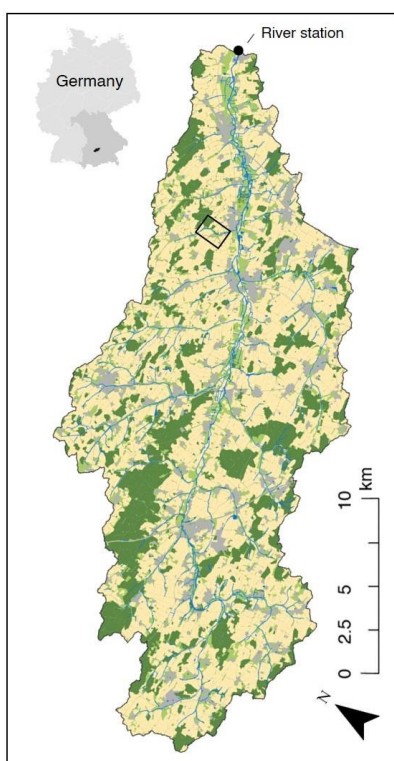

**Figure 1: The Glonn catchment (390 km²) representing a typical intensively used arable landscape**
**in Southern Germany. The black rectangle within the catchment marks the section of the detailed**
**maps in Fig. 7.**



*2.2. Model*

The erosion and MP transport is modelled based on a modified version of the spatially distributed water and tillage erosion and carbon (C) turnover model SPEROS-C (Fiener et al., 2015; Van Oost et al., 2005a). The model was originally developed to analyse the long-term effect of soil erosion on landscape-scale carbon balance (e.g. Nadeu et al., 2015), whereas the erosion components are based on the erosion and sediment transport model WaTEM/SEDEM, which was extensively tested and validated in different regions of the world (Krasa et al., 2005; Van Oost et al., 2000; Van Rompaey et al., 2001; Verstraeten and Prosser, 2008). The most important model components for this study are: (i) the water erosion and sediment transport component, (ii) the tillage erosion component, and (iii) the lateral redistribution and the vertical mixing of MP in the soil profile following erosion and deposition processes. As the C turnover component of SPEROS-C was not used in this study but the MP component was introduced, the model will subsequently be referred to as SPEROS-MP.

*Water erosion component:* The water erosion component of SPEROS-MP consists of two main parts. First, the erosion potential of each raster cell (5 m x 5 m) is estimated based on the German version of the Universal Soil Loss Equation ABAG (Schwertmann et al., 1987). The major advantage of this well-tested approach is that the input data to calculate the different USLE (ABAG) factors are available from the Bavarian State Office of Agriculture (Bayerische Landesanstalt für Landwirtschaft; LfL) and are regularly updated by the State Office administration. Sediment transport per raster cell, and hence deposition if transport capacity is smaller than sediment influx, is calculated using Eq. 1:

$$T_c = k_{tc} \cdot R \cdot C \cdot K \cdot LS_{2D} \cdot P \tag{Eq. 1}$$

Where $T_c$ is the transport capacity (kg m$^{-1}$ a$^{-1}$), $k_{tc}$ is the transport coefficient; $R$ (N h$^{-1}$ a$^{-1}$), $C$ (-), $K$ (kg h m$^{-2}$ N$^{-1}$) and $P$ (-) are the rainfall erosivity, soil cover, soil erodibility, and management factors of the



USLE calculated for Bavaria following the approach of Fiener et al. (2020. $LS_{2D}$ is a grid cell-specific
topographic combined slope gradient and lengths factor calculated following Desmet and Govers (1996,
using the digital elevation model (DEM) with a resolution of 5 m x 5 m.
*Tillage erosion component:* Tillage erosion is calculated based on a diffusion-type equation adopted
from (Govers et al., 1994)), which generally assumes that tillage erosion is proportional to slope gradient.
Consequently, tillage erosion or deposition is most prominent if slope gradient changes, with most soil
loss modelled at convexities and most soil accumulation at concavities. Tillage erosion has no direct
effect on sediment or MP delivery into the stream network, but over time it modifies the MP
concentration in the plough layer of different raster cells, leading to a decrease in MP delivery, because
at erosional sites subsoil with little potential MP is mixed into the plough layer, while MP at depositional
sites is buried below the plough layer.
*MP redistribution and vertical mixing:* It is generally assumed that MP is entering the soil via its
surface and is immediately mixed into the plough layer (upper 0.2 m). The MP input to arable land is
estimated at field level (see input estimate below). For MP erosion the concentration in the plough layer
of each 5 m x 5 m raster cell was multiplied with the bulk soil erosion of this raster cell to calculate the
MP outflux to neighbouring cells. The MP concentration of the transported sediment is analogously used
to calculate potential MP deposition. After each year of modelling water and tillage erosion, the soil
profile is updated assuming a tillage operation to a constant depth of 0.2 m. Consequently, MP-free
subsoil is mixed into the plough layer at erosional sites, decreasing the topsoil MP concentration, while
at depositional sites the deposited MP is mixed with the underlying old plough layer, creating a new
topsoil MP concentration and some MP in the layer no longer reached by the plough. Over the years this
creates a steadily increasing variability in MP concentration within fields and transports MP into soils of
other land uses (e.g. grassland and forest sites) assumed not to get other MP inputs.



*2.3. Data*

*2.3.1.Soil erosion model inputs and parameters*

For the study area, the LfL provided a digital elevation model (DEM, raster 5 m x 5 m), land-use data
(field based) and a soil map (1:25,000) as well as most USLE factors (Tab. 1). A transport capacity
coefficient $k_{tc}$ of 150 m was used as the optimum value for cropland for a 5 m x 5 m grid resolution
(Dlugoß et al., 2012). For the sake of simplicity and because long-term data on soil management was
missing, only the rainfall erosivity (*R* factor of the USLE) was calculated on a yearly basis, following
the approach of Schwertmann et al. (1987, using the mean annual precipitation N (mm/a). N was
available in a 1 km x 1 km grid resolution from the German Weather Service (DWD, 2020). We assumed
a corn-grain crop rotation (with a mixture of small grain crops and a proportion of row crops of 25%)
typically found in the region and used the USLE calculator developed by Brandhuber et al. (2018,
resulting in a *C* factor of 0.15, which is constantly used for all arable land in the catchment (Tab. 1). In
the case of forest and grassland, a low *C* factor of 0.004 and for settlements a *C* factor of 0.001 was
applied (Brandhuber et al., 2018). A *K* factor map was provided by the LfL (derived from the soil
properties given by the soil overview map of Bavaria at a scale of 1:25,000) based on the calculation in
Schwertmann et al. (1987. The $LS_{2D}$ factor was derived from the 5 m x 5 m DEM, following the approach
of Desmet and Govers (1996. Assuming some soil conservation methods to be in place, e.g. partial
contour ploughing, the *P* factor was set to 0.85 (Fiener et al., 2020). The tillage transport coefficient $k_{til}$
depends on the tillage implement, tillage speed, tillage depths, bulk density, texture and soil moisture at
time of tillage (Van Oost et al., 2006). For the tillage erosion modelled, a constant $k_{til}$ value of 350 kg m$^{-}$
$^{1}$ a$^{-1}$ for all fields was assumed (Tab. 1), which is a conservative estimate of a mixture of mouldboard
and chisel ploughing (Van Oost et al., 2006).



**Table 1: USLE factors used in SPEROS-MP.**

| Factors of the USLE | Value | Unit | Comment | Reference |
|---|---|---|---|---|
| $k_{tc}$ | 150 | m | | *Dlugoß et al., 2012* |
| $R$ | 0.048-0.089 | N h$^{-1}$ a$^{-1}$ | Varies annually, **controls the variability of the model** | *DWD (2020* |
| $C$ | | | | |
| *Arable land* | 0.15 | - | Does not vary spatially within different land uses | *Brandhuber et al., 2018* |
| *Forest and grassland* | 0.004 | - | | |
| *Urban area* | 0.001 | - | | |
| $K$ | 5-55 | kg h m$^{-2}$ N$^{-1}$ | Varies spatially depending on soil texture | *Fiener et al., 2020* |
| $P$ | 0.85 | - | | *Fiener et al., 2020* |
| $k_{til}$ | 350 | kg m$^{-1}$ a$^{-1}$ | | *Van Oost et al. 2006* |


*2.3.2.MP contamination of soils*
Because sampling and sample analysis would be extremely time consuming and costly, it is not
possible to determine the actual MP concentrations in a 390 km² catchment where estimates from MP
inputs suggest large spatial heterogeneity. Hence, the potential soil-MP contamination needs to be
estimated from the potential MP input from different sources. As soil erosion is dominant on arable land,
an exclusive input estimate was performed for arable land. However, it is important to emphasize that
most estimates are based on regional means for the whole of Bavaria and that any estimates of the MP
accumulated in the catchment soils since the 1950s are based on a number of assumptions and
simplifications, resulting in large uncertainties. To account for these uncertainties in the model outputs
and arrive at a robust indication of the potential contribution of soil erosion as a source of MP in the
stream network, we estimated the potential yearly mean, minimum and maximum soil-MP input for each
input pathway (see below) and did separate and combined modelling runs for the different contamination
estimates. As mentioned earlier, mean MP inputs from sewage sludge, compost and atmospheric
deposition were estimated from means for all arable land in Bavaria, while input of tyre wear was derived



using catchment specific road data and road specific traffic data as far as possible. These represent the
typical sources in the agricultural landscape of Southern Germany, along with MP, applicable for
SPEROS-MP. Other potential MP input pathways, for instance from plastic used in agricultural
management (e.g. mulch films) or from littering, were not considered for two main reasons. (i) In Bavaria
mulch films are mostly associated with certain regions where specific crops or vegetables are grown,
especially asparagus. For our test site this is not the case, and using the average area of mulch cover in
Bavaria to estimate the potential mean input in the catchment would have resulted in very small input
amounts, not comparable with other regions in the world, where mulch films can be a very important
source of MP (Li et al., 2022; Liu et al., 2014). (ii) Larger macroplastic fragments from mulch films and
littering should only be transported with severe rill and ephemeral gully erosion, which are not the
dominant erosion processes in the region.

*2.3.3. Sewage sludge and compost*

Sewage sludge and compost as soil amendments (organic fertilizers) contain different quantities of

microplastic and, in the case of compost, small macroplastic. The first step was to estimate the amount
of sewage sludge and compost applied on Bavarian agricultural soils since 1950. Bavarian waste reports
(LfU, 1990-2020) allowed us to determine the mean annual input on arable land for the time period
1990–2020. Historical application rates of compost were determined based on a linear relationship
between application rates and population numbers between 1990 and 2020 (the variability was continued
at random) (LfStaD, 2022) (Fig. 2b, c). In the case of sewage sludge, the number of residents connected
to the sewage system was taken into account (Schleypen, 2017). The gaps between historical individual
values were interpolated. The development of plant technology and the use of sewage sludge between
1945 and 1990 were considered, as described by Schleypen (2017. While compost was constantly used
as an organic fertilizer, the use of sewage sludge was quite variable over time (Fig. 2c). From 1970



onwards new wastewater treatment plant (WWTP) technology meant that the sewage sludge was no
longer allowed to accumulate dry, but rather as wet sludge (Schleypen, 2017). This led to a sharp drop
in the use of sewage sludge as a fertilizer and it was not until the 1990s that it become popular again
(Fig. 2c). Since 2017, the application of sewage sludge has been largely banned in Bavaria (Schleypen,

2017).

The second step was to estimate the MP concentrations in sewage sludge and compost. To do this,
current literature values were used to estimate the MP concentrations for 2020. A minimum, mean and
maximum MP concentration was always considered, based on the range of values from literature. For
sewage sludge, data from Edo et al. (2020 were used; this is, to our knowledge, one of the few studies
providing a mass balance of MP for a WWTP by specifying the total wastewater volume and the total
amount of sewage sludge per day. The sum of the MP particles filtered out (contained in sewage sludge)
and the delivered MP from the WWTP effluent results in the number of MP detected in the WWTP input.
Edo et al. (2020 consider size classes 25–104 µm, 104–375 µm and 375–5,000 µm and their data show
that 95% of the MP in the WWTP is retained in the sewage sludge, which is consistent with other
publications giving ranges of 93–98% (Habib et al., 2020; Tang and Hadibarata, 2021; Unice et al.,
2019). For compost, data from Braun et al. (2021 were used, which contain all essential data on MP in
compost from Germany. They examined MP in the size ranges < 1,000 µm, 1,000–5,000 µm and > 5,000
µm. For the mass calculation of the MP in compost, macroplastics are also included.



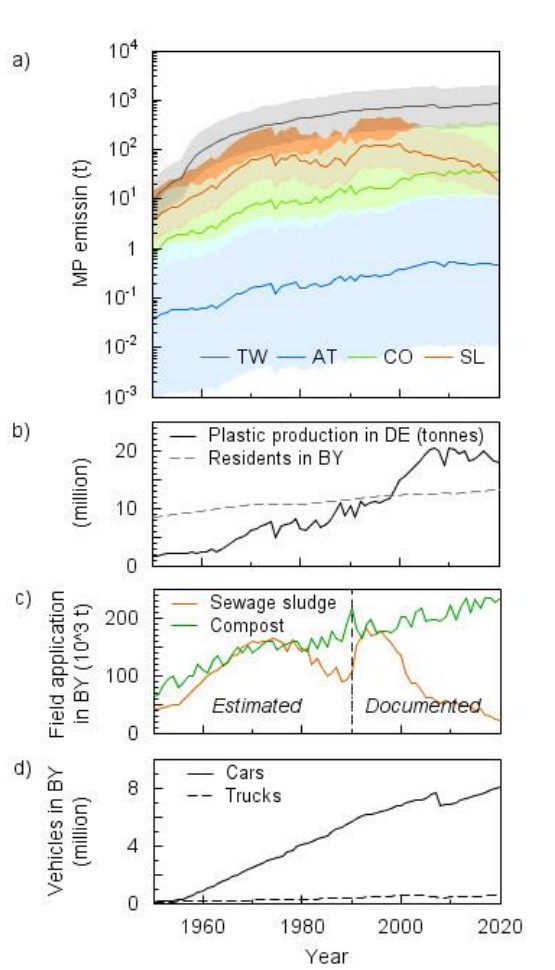

**Figure 2: a) The MP emissions for arable land in Bavaria from the different sources, tyre wear**
**(TW), sewage sludge (SL), compost (CO) and atmospheric deposition (AT), from 1950 to 2020. b)**
**The development of plastics production in Germany and the population of Bavaria since 1950. c)**
**Amount of application of sewage sludge and compost as fertilizer on Bavarian arable land. d)**
**The number of registered cars and trucks in Bavaria since 1950.**





Both publications, Edo et al. (2020 and Braun et al. (2021, provide information on the size distribution
of the detected MP particles. This enabled the most accurate conversion possible between mass and
particle number. When converting, the particle size, size distribution and shape were taken into account.
While a spherical shape was assumed for sewage sludge, for compost the most realistic possible volume
for each detected particle was calculated (individual dimensions have been provided by the authors of
Braun et al. (2021). Based on the type of plastic detected, an average density of 1 was assumed for all
particles. An average MP load of 1.14 g MP kg$^{-1}$ dry matter of sewage sludge (min.: 0.42 g, max.: 4.04
g) and 0.15 g MP kg$^{-1}$ dry matter of compost (min.: 0.05 g, max.: 1.36 g) was assumed.
Based on the known amounts of sewage sludge and compost applied, it was possible to calculate the
corresponding amount of MP that ends up on Bavarian agricultural soils (kg m$^{-2}$). When calculating the
MP concentration back to 1950, the amount of plastics produced in Germany was considered for each
year, as the MP concentration depends on the level of production (Fig. 2a, b). The annual amount of MP
was then evenly distributed across all agricultural fields in Bavaria, since spatial allocation within the
study area was not possible.
Between 1950 and 2020, a total of 7.26 million tonnes of sewage sludge and 11.7 million tonnes of
compost were added as organic fertilizer on agricultural fields in Bavaria. Hence it can be estimated that
4,090 t (min.: 1,510 t, max.: 14,500 t) and 1,110 t (min.: 358 t, max.: 10,100 t) of MP from sewage sludge
and compost, respectively, was dumped on arable land in Bavaria. From that, an average input on the
arable land in the Glonn River catchment of 42,100 kg MP from sewage sludge (min.: 15,500 kg, max.:
149,000 kg) and 11,500 kg MP from compost (min.: 3,660 kg, max.: 104,000 kg) was calculated. For
the arable land in the Glonn River catchment, this means an average annual MP application of 240 kg
MP from sewage sludge (min.: 90 kg, max.: 860 kg) and 370 kg from compost (min.: 120 kg, max.:



3,390 kg) in 2020 (Tab. 2). This results in a current entry rate of 1.14 mg MP m⁻² a⁻¹ (min.: 0.42 mg, 4.04
mg) from sewage sludge and 1.75 mg MP m⁻² a⁻¹ (min.: 0.56 mg, max.: 15.8 mg) from compost.
**Table 2: MP inputs into arable soils within the test catchment, separated by different sources. All**
**values are listed for the modelled time span 1950–2020 and separately for the year 2020.**

| | | Tyre wear | Sewage sludge | Compost | Atmospheric deposition | Unit |
|---|---|---|---|---|---|---|
| **1950–2020** | | | | | | |
| **MP application to arable land** | | **120,256** | **42,100** | **11,500** | **186** | **kg** |
| | min | 43,969 | 15,500 | 3,660 | 4.30 | |
| | max | 288,614 | 14,9000 | 104,000 | 4200 | |
| **2020** | | | | | | |
| **MP application to arable land** | | **3,109** | **240** | **370** | **4.76** | **kg** |
| | min | 1,137 | 90 | 120 | 0.11 | |
| | max | 7,462 | 860 | 3,390 | 107 | |
| **MP application rate** | | **19.67** | **1.14** | **1.75** | **0.02** | **mg MP m⁻² a⁻¹** |
| | min | 7.19 | 0.43 | 0.56 | 5*10-4 | |
| | max | 47.2 | 4.08 | 16.03 | 0.45 | |


*2.3.4. Atmospheric deposition*
For the atmospheric deposition of MP, the data from four bulk deposition measurements (precipitation
and dust deposition) in Bavaria (Witzig et al., 2021) were combined with the development of plastics
production in Germany since the 1950s. As no better data were available it was assumed that the
measured atmospheric deposition of MP in 2020 is proportional to German plastics production in general
(Fig. 2a). This results in a mean cumulative atmospheric MP input on arable land in Bavaria of 18 tons
of MP (min.: 0.41 t, max.: 407 t). Between 1950 and 2020, the arable land in the Glonn River catchment
was loaded with a total of 186 kg of MP (min.: 4.20 kg, max.: 4,200 kg). For 2020 an average annual
MP immission of 4.76 kg (min.: 0.11 kg, max.: 107 kg) or 0.02 mg MP m⁻² a⁻¹ (min.: 0.0005 mg, max.:
0.5 mg) via atmospheric deposition was calculated (Tab. 2).



*2.3.5.Tyre wear*

To determine the tyre wear particle input in the Glonn catchment we used existing traffic counting data from 2005, 2010 and 2015 for the main roads (motorways, federal roads, state roads and district roads) available from the Bavarian Road Information System (BAYSIS, 2015). Traffic volume for smaller roads (except farm roads) in rural areas were derived from a 1 km x 1 km population density grid following Gehrke et al. (2021. Based on these data the traffic volume (number of vehicles per km) for each paved road in the Glonn catchment could be estimated for the years 2005, 2010 and 2015. This was done separately for passenger cars (cars), heavy-duty vehicles (trucks) and motorcycles. For all other years, the traffic volume (number of vehicles per km) per road was linearly extrapolated based on the traffic volume in and the number of registered cars and trucks in Bavaria (LfStaD, 2022) (Fig. 2d). No emissions from unpaved roads and agricultural machinery were considered.

A minimum, medium and maximum scenario was considered, based on the quantity of released tyre particles specified in the literature. A mean tyre wear emission factor of 90 mg TW km$^{-1}$ (min.: 53 mg, max.: 200 mg) was assumed for cars (a motorcycle represents half a car) and 700 mg TW km$^{-1}$ (min.: 105 mg, max.: 1,7*10³ mg) for trucks, based on the reviews of Hillenbrand et al. (2005 and Wagner et al. (2018. Based on the length (km) and traffic volume (number of cars, motorbikes and trucks), the released TW was calculated for each section of road.

The transport of TW from roads into the surrounding soil systems was estimated based on literature information, assuming that the TW concentration exponentially declines with increasing distance from the road (Fig. 3). However, we could only identify one study (Müller et al., 2022) that directly measured TW contamination of soils with distance from the road, while most other studies (Motto et al., 1970; Werkenthin et al., 2014; Wheeler and Rolfe, 1979; Wik and Dave, 2009) used chemical markers and the distance from the road to estimate TW distribution. From all these different approaches we calculated a

median behaviour (Fig. 3). As the modelling is performed in a 5 m x 5 m grid, the land-use map may not
show all grass or vegetation strips often found along roads, which might lead to an overestimation of
TW input to arable land. Hence, we decided to use a conservative estimate, assuming that at least a 3 m
wide grass strip can be found on both sides of any road. Consequently about 85% of the TW produced
on any road (Fig. 3) cannot reach arable fields. The remaining 15% of TW that could potentially reach
arable land mostly settles within a 50 m distance from the road, whereas background MP concentrations
are reached in about 130 m distance (Fig. 3).

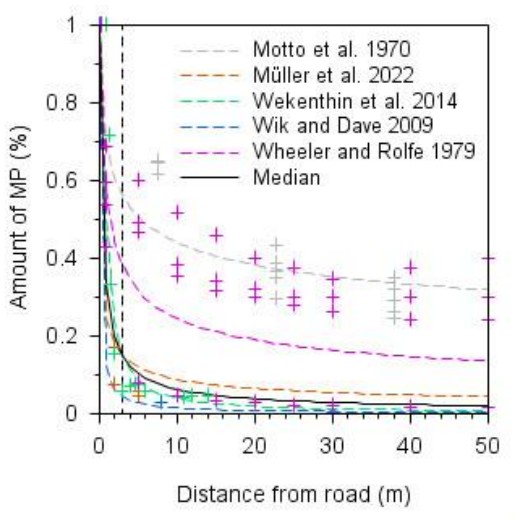

**Figure 3: The distribution of tyre wear in the soil relative to the distance from the road. Literature**
**values are based on direct detection of tyre wear (Müller *et al*. 2022) or on the estimated**
**concentrations of tyre wear particles based on chemical markers (Motto *et al*. 1970, Wheeler and**
**Dave 2009; Wik and Dave 2009; Wekenthin *et al*. 2014). The markers show the individual values,**
**the dashed lines show the mean of the respective reference. The black line represents the median**
**of all literature values used for modelling in this study.**

In comparison to the other MP sources considered (sewage sludge, compost and atmospheric deposition),
the estimate for TW was calculated on a field by field basis. To identify all agricultural fields affected





by road-borne TW deposits within a distance of 130 m, a land-use map was overlaid on the road network.
For each field, the area share of the associated road section and the distance to the road were considered
when calculating the TW load. The only limitation is that on fields affected by TW, in the model the
amount of TW was then distributed evenly over the entire field and not just on the affected field section
near the road (within 130 m).
Between 1950 and 2020, $120*10^3$ kg tyre wear (min.: $44*10^3$ kg, max.: $289*10^3$ kg) ended up on
arable land in the Glonn catchment (Tab. 2). In 2020 the average annual MP application amounts to
$3.1*10^3$ kg of tyre wear (min.: $1.1*10^3$ kg, max.: $7.5*10^3$ kg) (Tab. 2). The load from TW in 2020 can
reach maximum concentrations of $2.5*10^3$ mg TW $m^{-2}$ $a^{-1}$ on roads with heavy traffic use; the average
over all affected fields in the Glonn catchment area is 19.7 mg TW $m^{-2}$ $a^{-1}$ (Tab. 2).
*2.4. Model validation*
It is obviously impossible to validate the modelled MP delivery to the stream network against measured
MP loads, as this would call for a continuous monitoring of MP delivery for several years at least.
However, the modelled sediment delivery can be validated against measured data from the Bavarian
State Office for Environment (Bayerisches Landesamt für Umwelt, LfU), which operated a discharge
and sediment monitoring gauge in Hohenkammer (Fig. 1) between 1968 and 2020. At this gauge with a
defined river cross-section, daily discharge was derived from continuous runoff depth measurements in
combination with a stage discharge rating curve, while the stationarity of this rating curve at the
measuring cross-section was randomly checked once or twice every year. At the gauging station a weekly
water sample was collected (1968–2020) and its sediment concentration was determined in the
laboratory. From 2011 onwards a turbidity probe (Solitax ts-line; Hach Lange GmbH; Germany) was
installed and regularly calibrated against the samples taken by hand. Based on the continuous discharge





and the weekly to continuous sediment concentration measurements, the LfU provided daily sediment
load data for the time span 1968 to 2020, which were aggregated to yearly values for this study.
*2.5. Modelled scenarios*
Apart from modelling and analysing the MP delivery to the stream network via the erosion pathway
for the period from 1950 to 2020, we also modelled three scenarios (S1 to S3) to discuss potential future
pathways up to 2100.
*Scenario S1 – business-as-usual scenario:* In this scenario it is assumed that the MP input to arable
land continues until 2100 with the same input rates estimated for 2020. Given the ongoing increase in
plastics production (Chia et al., 2021; Lwanga et al., 2022), this may even be a conservative estimate of
a business-as-usual scenario pathway.
*Scenario S2 – spatially targeted application of soil amendments:* This scenario addresses two aspects.
(i) A potential reduction of MP delivery to the stream network due to a targeted application of soil
amendments, keeping a distance of at least 100 m from the stream network in the case of compost and
sewage sludge application. (ii) More generally illustrating the sensitivity of MP delivery to the stream
network in the case of non-homogenous MP inputs in the catchment. For the latter, soil amendments
were solely applied in the vicinity of the stream network (max distance 100 m).
*Scenario S3 – stop MP input:* This scenario is set up to determine the extent to which soils function
as a long-term source for MP with regard to soil erosion, assuming the MP applied before 2020 remains
stable in the soil until 2100. Therefore, a potential decline in MP concentration in the plough layer either
results from a lateral loss to neighbouring land uses (grassland or forest) or the stream network, or is
buried below the plough layer due to deposition processes (here deposition due to water and tillage
erosion).





**3. Results**
*3.1. Sediment delivery*
Without any calibration, the model satisfactorily reproduced the measured long-term mean sediment
delivery of the Glonn outlet (Fig. 4). The modelled sediment deliveries resulted in a mean of 145±18 kg
ha$^{-1}$, the measured mean contained 149±63 kg ha$^{-1}$ kg ha$^{-1}$ (Fig. 4). The model was obviously not able to
capture the full variability in the measured yearly sediment delivery (R² = 0.51; Fig. 4). It underestimates
years with high erosion rates, while it overestimates years with low erosion rates. However, we conclude
that the model performance (especially in reproducing the long-term mean) gives a solid basis for
modelling lateral MP fluxes due to erosion processes. Here it is important to note that our modelling
approach aims to estimate the magnitude of the MP erosion transport pathway, which was not analysed
in earlier studies, and that the estimated MP inputs contribute significantly to model uncertainty.

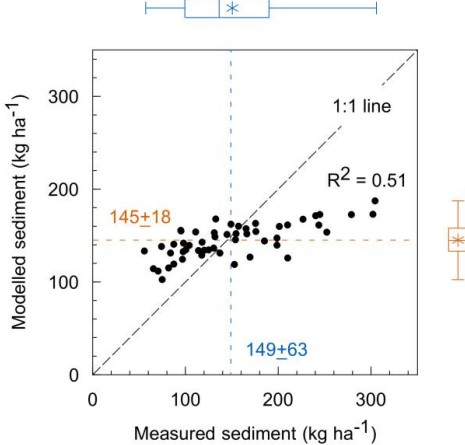


**Figure 4: Measured and modelled sediment delivery (1968 to 2020) at the outlet of the Glonn**
**catchment. The blue and orange lines represent the measured and modelled means, respectively.**
**The boxplots show the variability of the data. They show the median (line) and mean (star) and**
**the 1st and 3rd quartile, whiskers give the minimum and maximum.**



*3.2. MP erosion and delivery to stream network*

The constantly rising MP input to arable soils from different sources (Fig. 2) since 1950 is reflected in the steadily increasing, erosion-induced MP delivery into the stream network (Fig. 5a). Due to the long-term fertilization of arable land with sewage sludge, on average 0.51 kg of MP $a^{-1}$ entered the Glonn stream network in 2020 (Tab. 3). For compost it is 0.77 kg of MP $a^{-1}$, with 0.01 kg of MP $a^{-1}$ from atmospheric deposition (Tab. 3, Fig. 5a). With compost, sewage sludge and atmospheric deposition as potential MP inputs to arable land, SPEROS-MP generated a total MP input into the stream network of 1.29 kg MP via the soil erosion pathway in 2020. Deliveries to the stream network have also steadily increased in terms of TW (Fig. 5a), with an average 5.04 kg of MP $a^{-1}$ delivered to the stream network in 2020 (Tab. 3).



**Table 3: Soil erosion-induced MP delivery to the Glonn stream network, as well as redistribution**
**to grassland and forest. The MP vertical loss below the plough layer is also given. All values are**
**listed for the modelled time span 1950–2020 and separately for the year 2020.**

| | Tyre wear | Sewage sludge | Compost | Atmospheric deposition | Unit |
|---|---|---|---|---|---|
| **1950–2020** | | | | | |
| **MP delivery into stream network** | **134** | **57** | **17** | **0.32** | **kg** |
| min | 49.0 | 21 | 5 | 0.01 | |
| max | 322 | 200 | 155 | 9 | |
| *Percentage of MP application* | *0.11* | *0.14* | *0.15* | *0.17* | *%* |
| **MP delivery into grassland** | **604** | **442** | **82** | **1.5** | **kg** |
| min | 221 | 163 | 24 | 0 | |
| max | 1,450 | 1,551 | 748 | 42 | |
| *Percentage of MP application* | *0.50* | *1.05* | *0.71* | *0.81* | *%* |
| **MP delivery into forest** | **108** | **97** | **18** | **0.34** | **kg** |
| min | 39.5 | 36 | 5 | 0 | |
| max | 259 | 340 | 164 | 10 | |
| *Percentage of MP application* | *0.09* | *0.23* | *0.16* | *0.18* | *%* |
| **MP loss below plough layer** | **4,703** | **2605** | **489** | **14.8** | **kg** |
| min | 1,720 | 961 | 144 | 6 | |
| max | 11,287 | 9,414 | 4,458 | 386 | |
| *Percentage of MP application* | *3.91* | *6.19* | *4.25* | *8* | *%* |
| **2020** | | | | | |
| **MP delivery into stream network** | **5.04** | **0.51** | **0.77** | **0.01** | **kg MP a$^{-1}$** |
| min | 1.84 | 0.2 | 0.2 | 0.0003 | |
| max | 12.1 | 1.8 | 7 | 0.3 | |



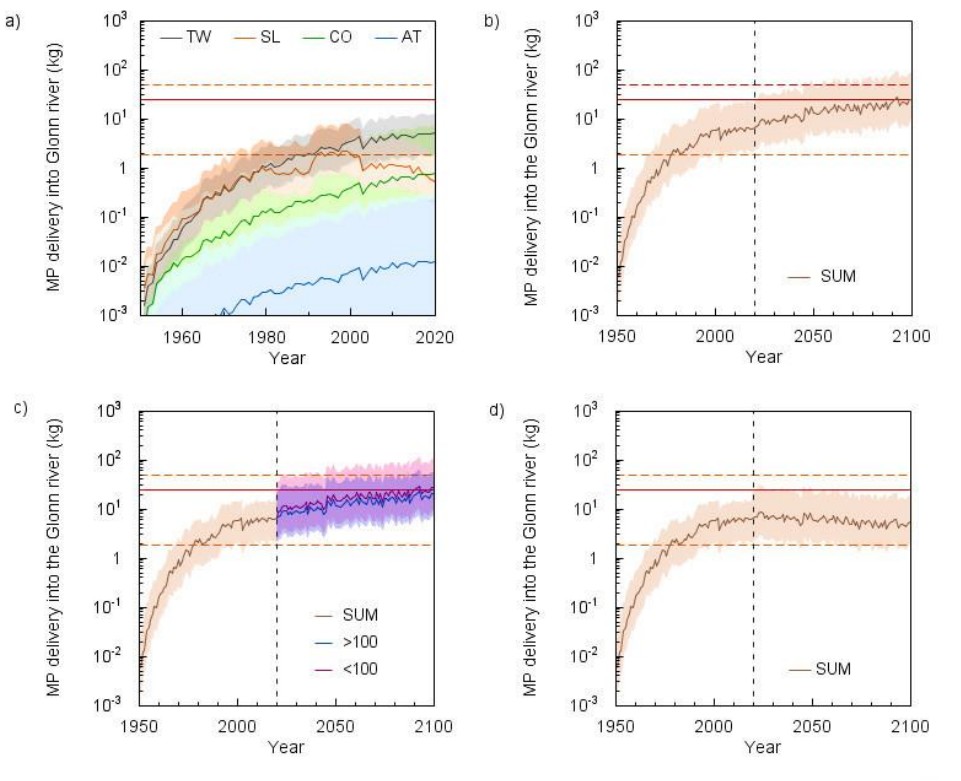

**Figure 5: MP delivery into the Glonn shown individually for tyre wear (TW), sewage sludge (SL), compost (CO) and atmospheric deposition (AT) or the sum of TW, SL, CO and AT (SUM). The dashed line gives the year 2020 as the starting point for different scenarios. For comparison, the amount of MP delivery through wastewater treatment plants (WWTP) in 2020 is shown as a red line (min. and max. as dotted lines). a) MP delivery into the Glonn river between 1950 and 2020. b) Result of scenario S1 with the assumption that the MP input will continue as in 2020. For comparison, the amount of MP delivery through wastewater treatment plants (WWTP) in 2020. c) Result of scenario S2. Compost and sewage sludge are applied to arable land at a distance of > 100 m and < 100 m from water streams. d) Result of scenario S3 with no MP input at all from 2020 onwards.**



Between 1950 and 2020, 208.3 kg of MP (134 kg TW, 57 kg sewage sludge, 17 kg compost and 0.32
kg atmospheric deposition) entered the Glonn stream network (Tab. 3), while overall a sediment load of
$3.0*10^8$ kg was delivered to the catchment outlet. TW was the main MP source, accounting for 64.3%,
followed by sewage sludge with 27.4%, compost with 8.2% and atmospheric deposition with 0.1%.
Taking into account the MP delivery relative to the MP input (i.e. total amount of MP input into soil in
1950–2020 vs. total MP delivery into the stream network from 1950–2020), only 0.14% of the MP
released to arable land was transported into the Glonn stream network. This differs slightly for the
different MP sources, ranging from 0.17% for atmospheric deposition to 0.11% for tyre wear (Tab. 3).
The spatially distributed model also allowed us to quantify the relocation of MP between different
land uses (an example is shown in Fig. 6f). The amount of MP delivered between 1950 and 2020 from
arable land to grassland and forest is $1.1*10^3$ and $0.2*10^3$ kg, respectively (Table 3). The larger delivery
to grasslands is particularly interesting, as these are mostly located along the stream network (see
discussion).
SPEROS-MP not only gives information about the MP relocation between arable land and other land
uses. The model also determines the amount of MP allocated below the plough layer (and thus out of
reach of water erosion) at depositional sites (an example is shown in Fig. 6e). Between 1950 and 2020,
3.9% of the TW supplied to arable land was moved below the plough layer (Tab. 3). This corresponds
to $4.7*10^3$ kg MP or 35 times the amount reaching the stream network via water erosion. For sewage
sludge it is 6.19% ($2.6*10^3$ kg), for compost 4.25% (489 kg) and for atmospheric deposition 8% (14.8
kg). Consequently, much more MP was translocated into the subsoil than was transported into the Glonn.
This transport into the subsoil was caused by water erosion (48.5%) and tillage erosion (51.5%).
Conversely, up to 95% of the MP applied to arable soil over the past 70 years remains in the plough layer
(infiltration and bioturbation excluded).

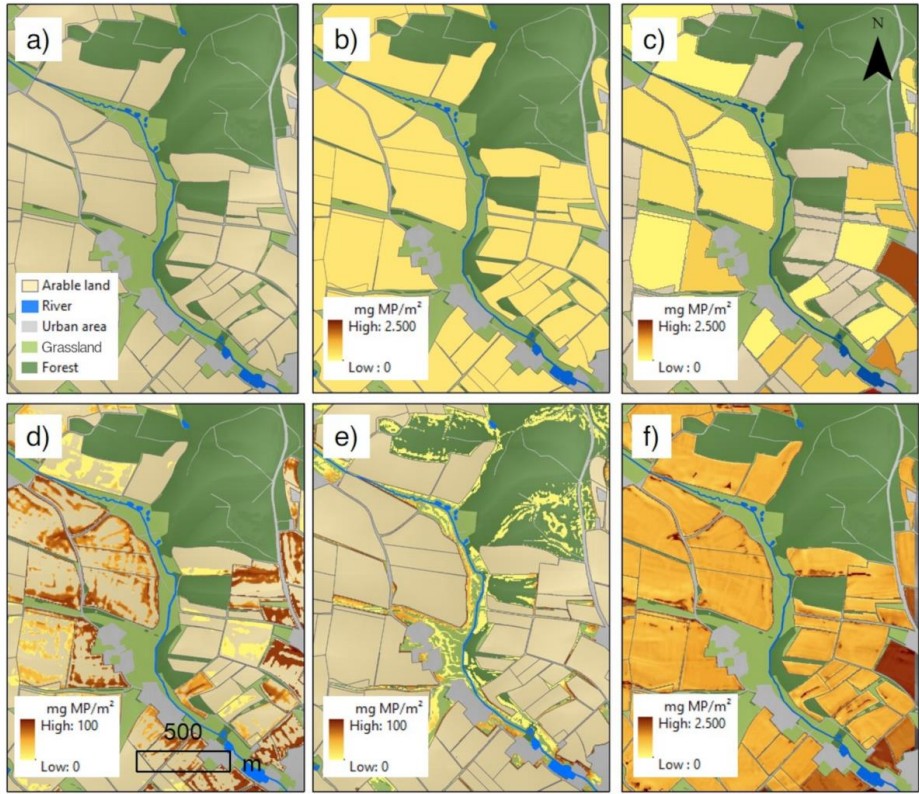

**Figure 6: Example of catchment segment (for location see Figure 1) illustrating microplastic (MP) input on arable land and results of erosion modelling between 1950 and 2020. The maps show the situation in 2020. a) Field-based land use. b) Total MP input from sewage sludge, compost and atmospheric input (without TW) as mean value over all arable land. c) MP input from TW, spatially distributed to individual arable fields. d) MP concentration below plough layer. e) MP transported to other land uses via soil erosion. f) MP distribution after water and tillage erosion on arable land. (DEM © Bayerische Vermessungsverwaltung)**



*3.3. Scenario S1 – business-as-usual*
If arable soils continue to be loaded with MP the same as in 2020, the annual MP delivery rate into the
Glonn stream network will increase by a factor of 4 by 2100. In 2100, 25.2 kg MP $a^{-1}$ (min.: 9.03 kg;
max.: 84.1 kg) through TW, compost, sewage sludge and atmospheric deposition would end up in the
stream network (Fig. 5b). Between 1950 and 2100, this would make a total MP input of 1.32 $*10^3$ kg MP
(min.: 511 kg; max.: 4.7 $*10^3$ kg) into the stream network.
*3.4. Scenario S2 – spatially targeted application of soil amendments*
In S2 MP inputs from atmospheric deposition and TW accumulation continued like in S1. However, the
location where the organic fertilizer (sewage sludge and compost) was applied in the catchment was
changed. All organic fertilizers were either applied at a distance of at least 100 m from the stream network
or within a distance smaller than 100 m along the stream network.
With an application at a distance of > 100 m, the MP delivery in the stream network would be reduced
to a total of 21.2 kg (min.: 7.72 kg; max.: 55.9 kg) in 2100 (Fig. 5c). That would correspond to a reduction
of 16% compared to S1. In the case of application at a distance of < 100 m, on the other hand, it would
be 27.9 kg (min.: 10 kg; max.: 102 kg) in 2100 and thus an increase of 10.7% compared to S1 (Fig. 5c).
The result becomes clearer if we consider TW and the organic fertilizers separately. If the distance is >
100 m, the annual MP delivery rate from organic fertilizer (sewage sludge and compost) without TW is
1.1 kg MP $a^{-1}$ (min.: 0.4 kg, max.: 7.8 kg) in 2100 (Fig. 7). For 2100, this would result in a 78% reduction
of the annual MP delivery rate from organic fertilizer into water bodies compared to S1. In total from





1950 to 2100, 173 kg MP (min.: 60 kg; max.: 1.0*10³ kg), or 46% less MP, from organic fertilizer would
end up in the stream network until 2100 (the effect of atmospheric input is negligible).
If organic fertilizer is applied along the stream network (max. distance < 100 m), a MP delivery of 7.8
kg a$^{-1}$ (min.: 2.6 kg, max.: 54 kg) is modelled in 2100 (Fig. 7). Between 1950 and 2100 a total of 493 kg
MP (min.: 168 kg; max.: 3.25*10³ kg) would be delivered to the river system by organic fertilizer
(without TW).

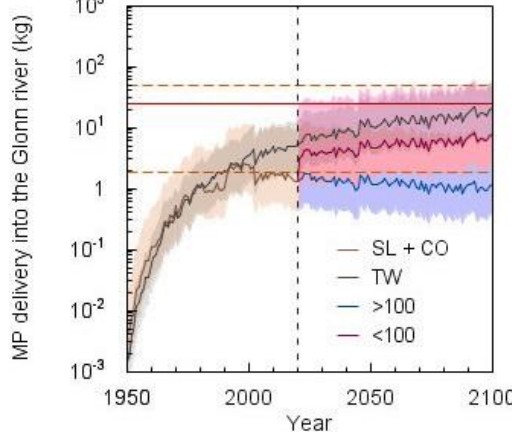

**Figure 7: Result of scenario S2 individually shown for tyre wear (TW) and for sewage sludge**
**(SL) plus compost (CO) together as organic fertilizer applied to arable land at a distance of > 100**
**m and < 100 m from water streams. For comparison, the amount of MP delivery through**
**wastewater treatment plants (WWTP) in 2020 as red lines (min and max as dotted lines).**

3.5. *Scenario S3 – stop MP input*:
In scenario S3 MP input stops from 2020 onwards. This abrupt stop in plastic immission is not reflected
in the MP delivery rates after 2020 (Fig. 5d). However, in the year 2100, 5.43 kg of MP a$^{-1}$ (min.: 1.98
kg, max.: 18.2 kg) would still end up in the stream network from arable land due to soil erosion (Fig.





5d). This corresponds to a decrease in the annual MP delivery rate of 14% between 2020 and 2100, with
80 MP-free years (since 2020). Since 1950, a total of 684 kg MP (min.: 246 kg; max.: 2*10³ kg) would
have ended up in the Glonn stream network.

**4.  Discussion**

*4.1. Modelled erosion rates (sediment delivery)*

The modelling approach used, with a yearly time step and the missing temporal and spatial variability

of most model input data (especially the constant crop cover factor), while only varying yearly rainfall
erosivity, leads to model outputs that do not capture the full temporal dynamics of the measured yearly
sediment delivery (Fig. 4). It is well documented that averaging model input variables over space and
time generally leads to the overestimation of years with low sediment delivery and underestimation of
years with high sediment delivery (Keller et al., 2021; Meinen and Robinson, 2021). The reduced
temporal variability in modelled sediment delivery is expected for two main reasons: (1) the annual
model time step averages out years where individual extreme events dominate the yearly sediment
delivery, and (2) varying only the annual rainfall erosivity, while all other input parameters (especially
cropping dynamics) are kept constant, cannot capture the temporal dynamics. However, without any
model calibration the model almost perfectly reflects the long-term mean sediment delivery between
1968 and 2020 (Fig. 4), explaining 51% of the variability in the measured data. Hence, we conclude that
SPEROS-MP is robust enough for this modelling study which focusses on MP delivery to the stream
network in the Glonn catchment, especially as uncertainties associated with the erosion modelling are in
any case smaller than the uncertainties associated with estimates of MP immissions to the arable soils in
the catchment.



*4.2. Plausibility of MP soil input estimates*
Estimating the cumulative MP-soil immissions from different sources for a period starting from 1950
is of course associated with large uncertainties. To account for these uncertainties, we deliberately used
large ranges of possible inputs in our semi-virtual catchment approach which in the following discussion
are compared with literature values for Germany or Bavaria as a whole.
4.2.1.*MP from sewage sludge, compost and atmospheric deposition*
Brandes et al. (2021 calculated mean MP inputs into agricultural soils in Germany for compost (1990–
2016) and for sewage sludge (1983–2016). For Bavaria, their calculation results in compost-MP input
rates of between 15 and 80 mg MP m$^{-2}$ a$^{-1}$ and sewage sludge-MP input rates between 0 and 190 mg MP
m$^{-2}$ a$^{-1}$. Bertling et al. (2021 also determined MP immissions (TW excluded) to agricultural soils in
Germany, resulting in much higher input rates for 2021 for compost and sewage sludge, with up to 702
mg MP m$^{-2}$ a$^{-1}$ and 2.1*10$^3$ mg MP m$^{-2}$ a$^{-1}$, respectively. In contrast to the first authors, Braun et al. (2021
calculate the possible MP load for the legally permissible amount of compost applied to fields in
Germany. This maximum permissible amount of compost application results in maximum possible entry
rates ranging from 34 to 4.7*10$^3$ mg MP m$^{-2}$ a$^{-1}$ into agricultural soils via compost.
For this study, an MP emission to arable soils of between 0.42 and 4 mg MP m$^{-2}$ a$^{-1}$ for sewage sludge
and between 0.56 and 15.8 mg MP m$^{-2}$ a$^{-1}$ for compost were calculated for Bavaria. Our values are not
based on the maximum possible limits, but on the most realistic estimates possible. Therefore, our MP
loads remain well below the literature values. Nevertheless, the MP input from compost is likely to be
underestimated, based on optical detection of MP > 1 mm (Bläsing and Amelung, 2018; Braun et al.,
2021; Weithmann et al., 2018). Currently, much more compost (21*10$^7$ t in 2020) is spread on fields in
Bavaria than sewage sludge (24*10$^4$ t in 2020), causing higher MP emissions from compost (Fig. 2a).



This results from the reduction in sewage sludge application, which has been largely banned in Bavaria
since 2017 (Schleypen, 2017) (Fig. 2c). However, regional policy strategies regarding the use of sewage
sludge differ substantially within Germany, making comparisons within the country somewhat difficult
(Brandes et al., 2021).
For atmospheric deposition, an average of 771 and 395 MP particles $m^{-2}$ $d^{-1}$ were measured at rural
locations in London and Hamburg (Klein and Fischer, 2019; Wright et al., 2019). Brahney et al. (2020
show that airborne microplastic particles accumulate at minimum concentrations of 48±7 MP particles
$m^{-2}$ $d^{-1}$ even in the most isolated areas of the United States (national parks and national wilderness areas).
Even in Antarctic snow up to 29 MP particles per melted litre were found (Aves et al., 2022). In this
study, the values of Witzig et al. (2021 were used to estimate the MP contribution via atmospheric
deposition. They made MP measurements at different locations in Bavaria, ranging from 74±19 to
109±16 MP particles $m^{-2}$ $d^{-1}$. Even if the transfer of such particle numbers to mass inputs is associated
with additional uncertainties, these amounts are orders of magnitude smaller than the inputs from sewage
sludge and compost and hence less important.
*4.2.2.Tyre wear*
The large MP mass resulting from tyre wear is noticeable in both the TW input data and the TW
delivery rates into the stream network. With modelled mean TW delivery of 5 kg MP $a^{-1}$ in 2020 into the
river system, the equivalent of half a car tyre ends up as MP in the Glonn (flow length of 50 km) each
year. However, the calculated mean TW input to the Glonn catchment of 200 mg MP $m^{-2}$ in 2020 is in
same the range as the estimates in other studies. For example, annual values of between 180 and 370 mg
TW $m^{-2}$ were reported for Germany (Baensch-Baltruschat et al., 2020; Kocher et al., 2010; Wagner et
al., 2018). The modelled MP input (see Fig. 3) to arable land in the Glonn catchment was substantially
smaller, with a mean of 19.7 mg TW $m^{-2}$.



Most of the TW remains on the roads or in the immediate vicinity. Some of the TW is expected to be
transported directly into surface waters via runoff from the road. Baensch-Baltruschat et al. (2020
estimated that 12–20% of the tyre wear released on German roads ends up in surface water via road
runoff. The hydrological model estimates of Unice et al. (2019 indicated that 18% of released tyre wear
was transported to freshwater in the Seine River catchment. In comparison, focusing on erosion of MP
which was mixed into the plough layer, only 0.11% of the applied TW to arable soils from 1950 to 2020
reached the river system. Although TW is the largest source of entry in our study, the MP flow to the
stream network is overall a conservative estimate. This mostly results from our assumption that all roads
are surrounded by a 3 m grass buffer strip (even if this was not shown in the 5 m x 5 m land-use raster
map used), always trapping at least 85% of the TW emissions (Fig. 3). Yet even this conservative
assumption is associated with high uncertainties. The width of the grass strip between the road and the
field has an enormous impact on the MP emission. A 2 m wide buffer strip would still retain
approximately 80% and a 1 m wide buffer strip approximately 65% of the TW emission (Fig. 3). Without
any assumed grass buffer strips, the MP emission from TW would be 8 times higher. Ultimately, the
spatially distributed tyre wear is still associated with uncertainties. The level of TW emissions into the
environment (not just arable land) makes other MP sources almost negligible, especially in terms of MP
saving strategies.
Overall, it can be concluded that our estimates of MP input to the Glonn catchment are in the same
order of magnitude, or somewhat smaller, compared to most other studies, and hence should be more or
less reasonable, even if any estimates are associated with large uncertainties (e.g. extrapolating back to
1950; the small number of studies available for estimating MP concentrations in sewage sludge and
compost; errors when transferring particle numbers in particle mass etc.). However, an error in modelling
the MP delivery into the stream network of the test catchment most likely results from the fact that mean
application rates (sewage sludge, compost) for the whole of Bavaria were used (Fig. 6b), while only TW



input was calculated on a catchment-specific basis (Fig. 6c). Again, it is important to note that the Glonn
catchment was used as an exemplar to address and discuss the potential magnitude of the MP/soil erosion
pathway in such mesoscale catchments determined by arable land use.
*4.3. The modelled fate of MP*
As a mass-balanced model, SPEROS-MP calculates the MP input in mass (kg m²) and not in particle
numbers. Hence, the model does not consider the type, shape, density, size or chemical properties of the
MP particles from different MP sources. It thus treats the erodibility of MP from all input pathways
equally. However, it can be assumed that particle properties play a decisive role for the erosion-induced
lateral transport, as well as for the potential vertical transport. Small MP particles should be translocated
faster below the plough layer due to bioturbation and maybe infiltration (Li et al., 2021; Rehm et al.,
2021; Waldschläger and Schüttrumpf, 2020). A subsequent reduction in MP concentration in the plough
layer will also reduce MP erosion. On the other hand, smaller MP particles might more strongly interact
with soil organic or mineral particles, or might even be included in soil aggregates, hence are more likely
transported as bulk soil. For example, Rehm et al. (2021 were able to demonstrate in a long-term plot
experiment that smaller PE particles (53–100 µm) are less strongly enriched in delivered sediments
compared to larger PE particles (250–300 µm). Such behaviour might change again with increasing
particle size, because if particles transported with sheet flow are larger than the flow depths (mostly < 1
mm), transport in suspension is no longer possible.
In general, the potential decrease in topsoil MP concentration due to infiltration and bioturbation is
not accounted for in SPEROS-MP. Vertical MP transport via infiltration and bioturbation has been
widely discussed and partially observed in earlier studies, e.g. (Rillig et al., 2017), while earthworms
play an especially important role in directly transporting MP via digestion and excretion (Huerta Lwanga



et al., 2017) or in preparing preferential flow pathways for MP leaching (Yu *et al*., 2019). Ignoring these
processes of vertical movement below the plough layer will potentially lead to a slight overestimation of
the topsoil MP concentration in the modelling approach presented here.
SPEROS-MP not only delivers MP into the stream network, but also redistributes MP within the
catchment and within the soil profile. As arable land in the catchment is mostly found on the upper
slopes, and grassland in the flood plains, large amounts of MP are transported from arable land to
grassland (Tab. 3). No tillage takes place in grassland, leading to high MP concentration in the topsoil.
Along the main river in particular, grassland contaminated with MP (example shown in Fig. 6f) offers a
high potential for MP loss during flood events. In the flood plains, the groundwater level is regularly
close to the surface, hence the chance of MP leaching to the groundwater increases (Chia et al., 2021;
Singh and Bhagwat, 2022; Viaroli et al., 2022).
*4.4. Soil erosion as a potential MP source for inland waters*
Comparing the annual MP input to arable land and the annual MP loss through soil erosion indicates
that only a very small proportion ($\leq 0.17\%$ since 1950) is delivered to the stream network. The loss rate
of TW (0.11%) was the smallest compared to sewage sludge, compost and atmospheric deposition (Tab.
3). This is because the TW was not applied to all fields, but only to the fields next to a road. The low
percentage of input lost to the streams should not lead to the fallacy that MP transport via soil erosion is
negligibly small (Schell et al., 2022; Weber et al., 2022). This becomes clearer when comparing the MP
input from soil erosion with the MP input from wastewater treatment plants (WWTP) in the study area
(Fig. 5). Based on the known number and size of the WWTPs in the study area and MP loads in German
WWTPs from literature (Mintenig et al., 2014), the MP delivery into the Glonn through WWTP outlets
can be estimated at an average of 25 kg MP $a^{-1}$ (min.: 1.9 kg, max.: 49 kg) in 2020 (Fig. 5). These values
represent a maximum scenario since the calculations were based on the possible full capacities of the





WWTPs. Within the test catchment, the MP delivery into the stream network was 6.3 kg MP a$^{-1}$ (min.:
2.2 kg, max.: 21 kg) in 2020, but (S1, Fig. 5b) could reach 25.2 kg MP a$^{-1}$ (min.: 9 kg, max.: 84.3 kg) by
the end of the century (Fig. 5b).
Rehm et al. (2021 have shown that due to its low density, MP is preferentially eroded and and is
enriched by up to a factor of four in delivered sediments. These potential enrichment effects were not included
in SPEROS-MP. In addition, other MP input sources such as plastic used in agriculture (e.g. mulch films)
and littering were not considered in this study. In this respect, therefore, the modelled MP delivery is a
conservative estimate. Overall, our results are in line with other, larger-scale model estimates for the
Bavarian section of the Danube catchment, showing that the MP input via soil erosion into water bodies
in rural areas outweighs the MP input of WWTP outlets (Witzig et al., 2021). It should therefore not be
claimed that soil erosion for MP transport is negligible (Schell et al., 2022) while wastewater treatment
plants are treated as a major MP source for inland waters (Cai et al., 2022; Eibes and Gabel, 2022;
Murphy et al., 2016).
*4.5. The MP sink function of soil results in a long-term MP source*
Today's MP pollution of arable land represents a long-term MP source for inland waters. With the
model scenarios S1 and S3, this study was able to show that the MP discharge from arable soils into
inland waters via soil erosion will still affect many generations to come, even if MP entry into the
terrestrial environment could be avoided. Because of low MP loss rates (≤ 0.17%) via soil erosion and
the stability of conventional plastic materials over centuries (Ng et al., 2018), the MP particles
accumulate in the soil over the years. As most of the MP stays in the plough layer (Tab. 3), it is made
available to surface runoff and erosion processes on a regular basis. After 80 years without MP input in
S3, MP delivery from the soil decreased only by 14%. The MP concentration in the topsoil of arable land
decreases over time due to lateral MP loss into the stream network or into neighbouring grassland and



forest areas (example shown in Fig. 6f). The MP concentration in the topsoil also decreases since erosion
incorporates MP-free subsoil and, on the other hand, MP gets below the plough layer at depositional sites
(outside the range of water erosion). It is important to note that tillage erosion plays an important role,
as it supports the burial of MP below the plough layer (example shown in Fig. 6e).
S3 is reminiscent of other well-known environmental problems of long-term diffuse pollution, e.g.
with phosphorus (Daneshgar et al., 2018; Vaccari, 2009), where a pollutant accumulates in soils but
slowly find its way into inland waters through soil erosion. In this respect, it is important to note that it
will be easier to reduce MP inputs to stream networks coming from point sources, e.g. WWTP, whereas
the diffuse input will continue for centuries.
*4.6. Targeted application of MP-laden organic fertilizer*
The predicted increase in plastics production means that more MP inputs into the environment can be
expected in the future (Borrelle et al., 2020; Horton, 2022). Because of this, it is necessary to consider
what measures can be taken to reduce or avoid the entry of MP into the various environmental
compartments. The results of S2 have shown that the application of organic fertilizer (without TW)
containing MP at a distance of more than 100 m from the stream network can reduce MP entry into
surface waters via soil erosion by up to 46% compared to S1 (Fig. 7). By contrast (unplanned) application
of MP-laden soil amendments in the proximity of the stream network increase MP supply (by 53% in
our scenario).
This highlights the potential impact of optimized landscape management taking into account the
location of any agricultural management activity. It also shows that, in addition to soil conservation in
the field to prevent soil erosion, general changes in catchment management affecting hydrological and
sedimentological connectivity have important implications for the transport of sediments and pollutants.



658 Therefore, the location of soil additives, which are usually used to close production cycles, should be

659 considered for future use. This consideration can have a significant influence on the subsequent erosion

660 transport and redistribution of, for example, MP within a whole river catchment.

661 **5. Conclusion**

662 In this study, the transport of MP eroded from arable land was modelled across a mesoscale landscape.

663 Sewage sludge, compost, atmospheric deposition and tyre wear were considered as MP sources. Tyre

664 wear not only represented the largest MP input to arable land. It also generated the largest MP delivery

665 rates to the stream network — although tyre wear is not widespread on arable land, only occurring on

666 fields near the roads. In percentage terms, only a small fraction ($< 0.2\%$) of all MP applied to arable land

667 ended up directly in the stream network via soil erosion. However, the MP mass delivered into the stream

668 network represented a serious amount of MP input. The modelled MP delivery into the stream network

669 was in the same range of potential MP inputs from wastewater treatment plants from this rural area.

670 In addition, was shown that most of the MP applied to arable soils remains in the topsoil (0–20 cm)

671 for decades. Tillage produces a regular homogenization, and the MP stays available for surface runoff

672 and water erosion in the long term. Based on a series of scenarios modelled up to 2100 with no more MP

673 input from 2020 onwards, similar MP delivery rates (compared to 2020) could still be identified. This

674 implies that arable land represents an MP sink on the one hand and a long-term MP source for inland

675 waters on the other.

676 Using the soil profile update component included in the SPEROS-MP model, the MP concentrations

677 along the soil profile could be determined to a depth of 1 m. It was modelled that 5% of the MP applied

678 to arable land is translocated into the subsoil ($> 20$ cm) by tillage and water erosion. Located below the

679 plough horizon, the MP is out of reach for future lateral surface runoff erosion processes. Based on the



spatially distributed erosion model, it was also demonstrated that most of the eroded MP leaving arable
land is deposited in grassland (1% of applied MP). Especially in areas of the river valleys, these
accumulations could represent a concentrated MP entry into the stream network in the event of a flood.

The most effective protection for arable land would probably be to limit or ban the application of MP-

contaminated organic fertilizers. The following measures would be conceivable to protect water bodies
from MP inputs through soil erosion. Our model scenario showed that the targeted application of MP-
contaminated organic fertilizer at a distance of at least 100 m from the water body led to a significantly
lower MP delivery rate from this MP source. The deliberate creation of grass strips in the landscape to
protect against erosion would also be an option. However, it is important to consider that all calculated
and modelled cases were dominated by tyre wear, which is difficult to manage, especially in regions with
a high population and dense road network. Therefore, in order to preserve soil as a valuable resource, as
well as to protect the terrestrial and aquatic ecosystem from MP pollution and its effects, we should focus
on limiting MP emissions to the environment in general as much as possible.



**Competing interests**

Some authors are members of the editorial board of journal SOIL. The peer-review process was guided by an independent editor, and the authors have also no other competing interests to declare.

**Acknowledgments**

The authors would like to acknowledge the financial support from the Federal Ministry of Education and Research towards this research as part of the initiative Plastics in the Environment (funding number 02WPL1447A-G). In addition, we would like to thank the Bavarian State Office of Agriculture (LfL) and the Bavarian State Office for the Environment (LfU) for providing and accessing Bavaria-wide data, as well as providing the modelling data for the Glonn catchment area. Finally, special thanks go to the members of the Soil and Water Resources Research Group in Augsburg for supporting this work.



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

in Czech Republic, Austria, and Bavaria—Differences in Model Adaptation, Parametrization, and Data
Availability. Applied Sciences 2020; 10: 3647.
Fiener P, Govers G, Van Oost K. Evaluation of a dynamic multi-class sediment transport model in a catchment
under soil-conservation agriculture. Earth Surface Processes and Landforms 2008; 33: 1639-1660.
Fiener P, Wilken F, Aldana-Jague E, Deumlich D, Gómez J, Guzmán G, et al. Uncertainties in assessing tillage
erosion–how appropriate are our measuring techniques? Geomorphology 2018; 304: 214-225.
Gehrke I, Dresen B, Blömer J, Sommer H, Lindow F, Röckle R. TyreWearMapping. Digitales Planungs-und
Entscheidungsinstrument zur Verteilung, Ausbreitung und Quantifizierung von Reifenabrieb in
Deutschland. Schlussbericht. 2021.
Govers G, Vandaele K, Desmet P, Poesen J, Bunte K. The role of tillage in soil redistribution on hillslopes.
European Journal of Soil Science 1994; 45: 469-478.
Guo J-J, Huang X-P, Xiang L, Wang Y-Z, Li Y-W, Li H, et al. Source, migration and toxicology of microplastics
in soil. Environment International 2020; 137: 105263.
Habib RZ, Thiemann T, Al Kendi R. Microplastics and wastewater treatment plants—a review. Journal of Water
Resource and Protection 2020; 12: 1.
Heinze WM, Mitrano DM, Cornelis G. Bioturbation-driven transport of microplastic fibres in soil. Copernicus
Meetings, 2022.
Hillenbrand T, Toussaint D, Boehm E, Fuchs S, Scherer U, Rudolphi A, et al. Discharges of copper, zinc and lead
to water and soil. Analysis of the emission pathways and possible emission reduction measures; Eintraege
von Kuper, Zink und Blei in Gewaesser und Boeden. Analyse der Emissionspfade und moeglicher
Emissionsminderungsmassnahmen.  2005.
Horton AA. Plastic pollution: When do we know enough? Journal of Hazardous Materials 2022; 422: 126885.
Huerta Lwanga E, Thapa B, Yang X, Gertsen H, Salanki T, Geissen V, et al. Decay of low-density polyethylene
by bacteria extracted from earthworm's guts: A potential for soil restoration. Sci Total Environ 2017; 624:
753-757.
Hurley RR, Nizzetto L. Fate and occurrence of micro(nano)plastics in soils: Knowledge gaps and possible risks.
Current Opinion in Environmental Science & Health 2018; 1: 6-11.
Keller B, Centeri C, Szabó JA, Szalai Z, Jakab G. Comparison of the applicability of different soil erosion models
to predict soil erodibility factor and event soil losses on loess slopes in Hungary. Water 2021; 13: 3517.
Klein M, Fischer EK. Microplastic abundance in atmospheric deposition within the Metropolitan area of Hamburg,
Germany. Science of the Total Environment 2019; 685: 96-103.
Knight LJ, Parker-Jurd FN, Al-Sid-Cheikh M, Thompson RC. Tyre wear particles: an abundant yet widely
unreported microplastic? Environmental Science and Pollution Research 2020: 1-10.
Kocher B, Brose S, Feix J, Görg C, Peters A, Schenker K. Stoffeinträge in den Straßenseitenraum-Reifenabrieb.
BERICHTE  DER  BUNDESANSTALT  FUER  STRASSENWESEN.  UNTERREIHE
VERKEHRSTECHNIK 2010.
Krasa J, Dostal T, Van Rompaey A, Vaska J, Vrana K. Reservoirs' siltation measurments and sediment transport
assessment in the Czech Republic, the Vrchlice catchment study. Catena 2005; 64: 348-362.
LfStaD BLfSuD. Statistisches Jahrbuch für Bayern.  2022.
LfU BLfU. Abfallwirtschaft–Hausmüll in Bayern–Bilanzen 2002. Bayerisches Landesamt für Umweltschutz,
Augsburg 1990-2020.



Li H, Lu X, Wang S, Zheng B, Xu Y. Vertical migration of microplastics along soil profile under different crop
root systems. Environmental Pollution 2021; 278: 116833.
Li S, Ding F, Flury M, Wang Z, Xu L, Li S, et al. Macro-and microplastic accumulation in soil after 32 years of
plastic film mulching. Environmental Pollution 2022; 300: 118945.
Lian J, Liu W, Meng L, Wu J, Zeb A, Cheng L, et al. Effects of microplastics derived from polymer-coated fertilizer
on maize growth, rhizosphere, and soil properties. Journal of Cleaner Production 2021; 318: 128571.
Liu EK, He WQ, Yan CR. 'White revolution' to 'white pollution'—agricultural plastic film mulch in China.
Environmental Research Letters 2014; 9.
Lwanga EH, Beriot N, Corradini F, Silva V, Yang X, Baartman J, et al. Review of microplastic sources, transport
pathways and correlations with other soil stressors: a journey from agricultural sites into the environment.
Chemical and Biological Technologies in Agriculture 2022; 9: 1-20.
Meinen BU, Robinson DT. Agricultural erosion modelling: Evaluating USLE and WEPP field-scale erosion
estimates using UAV time-series data. Environmental Modelling & Software 2021; 137: 104962.
Mintenig S, Int-Veen I, Löder M, Gerdts G. Mikroplastik in ausgewählten Kläranlagen des Oldenburgisch-
Ostfriesischen Wasserverbandes (OOWV) in Niedersachsen. 2014.
Motto HL, Daines RH, Chilko DM, Motto CK. Lead in soils and plants: its relation to traffic volume and proximity
to highways. Environmental Science & Technology 1970; 4: 231-237.
Müller A, Kocher B, Altmann K, Braun U. Determination of tire wear markers in soil samples and their distribution
in a roadside soil. Chemosphere 2022; 294: 133653.
Murphy F, Ewins C, Carbonnier F, Quinn B. Wastewater Treatment Works (WwTW) as a Source of Microplastics
in the Aquatic Environment. Environ Sci Technol 2016; 50: 5800-8.
Nadeu E, Gobin A, Fiener P, Van Wesemael B, Van Oost K. Modelling the impact of agricultural management on
soil carbon stocks at the regional scale: the role of lateral fluxes. Global Change Biology 2015: DOI:
10.1111/gcb.12889.
Nasseri S, Azizi N. Occurrence and Fate of Microplastics in Freshwater Resources. Microplastic Pollution.
Springer, 2022, pp. 187-200.
Ng E-L, Lwanga EH, Eldridge SM, Johnston P, Hu H-W, Geissen V, et al. An overview of microplastic and
nanoplastic pollution in agroecosystems. Science of the total environment 2020; 627: 1377-1388.
Ng EL, Lwanga EH, Eldridge SM, Johnston P, Hu HW, Geissen V, et al. An overview of microplastic and
nanoplastic pollution in agroecosystems. Science of the Total Environment 2018; 627: 1377-1388.
Nunes JP, Wainwright J, Bielders CL, Darboux F, Fiener P, Finger D, et al. Better models are more effectively
connected models. Earth Surface Processes and Landforms 2018; 43.
Pérez-Reverón R, González-Sálamo J, Hernández-Sánchez C, González-Pleiter M, Hernández-Borges J, Díaz-Peña
FJ. Recycled wastewater as a potential source of microplastics in irrigated soils from an arid-insular
territory (Fuerteventura, Spain). Science of The Total Environment 2022; 817: 152830.
Rehm R, Zeyer T, Schmidt A, Fiener P. Soil erosion as transport pathway of microplastic from agriculture soils to
aquatic ecosystems. Science of The Total Environment 2021; 795: 148774.
Rillig MC, Ziersch L, Hempel S. Microplastic transport in soil by earthworms. Sci Rep 2017; 7: 1362.
Sajjad M, Huang Q, Khan S, Khan MA, Yin L, Wang J, et al. Microplastics in the soil environment: A critical
review. Environmental Technology & Innovation 2022: 102408.
Schell T, Hurley R, Buenaventura NT, Mauri PV, Nizzetto L, Rico A, et al. Fate of microplastics in agricultural
soils amended with sewage sludge: Is surface water runoff a relevant environmental pathway?
Environmental Pollution 2022; 293: 118520.
Scheurer M, Bigalke M. Microplastics in Swiss Floodplain Soils. Environmental science & technology 2018.
Schleypen P. Abwasserbehandlung (nach 1945). Historisches Lexikon Bayerns 2017.
Schmidt J, v.Werner M, Michael A. Application of the EROSION 3D model to the CATSOP watershed, The
Nederlands. Catena 1999; 37: 449-456.
Schwertmann U, Vogl W, Kainz M. Bodenerosion durch Wasser. Ulmer Verlag, 64 p 1987.
Singh S, Bhagwat A. Microplastics: A potential threat to groundwater resources. Groundwater for Sustainable
Development 2022: 100852.
Sommer F, Dietze V, Baum A, Sauer J, Gilge S, Maschowski C, et al. Tire abrasion as a major source of
microplastics in the environment. Aerosol and Air Quality Research 2018; 18: 2014-2028.



Tang KHD, Hadibarata T. Microplastics removal through water treatment plants: Its feasibility, efficiency, future
prospects and enhancement by proper waste management. Environmental Challenges 2021; 5: 100264.
Tian L, Jinjin C, Ji R, Ma Y, Yu X. Microplastics in agricultural soils: sources, effects, and their fate. Current
Opinion in Environmental Science & Health 2022; 25: 100311.
Unice KM, Weeber MP, Abramson MM, Reid RCD, van Gils JAG, Markus AA, et al. Characterizing export of
land-based microplastics to the estuary - Part I: Application of integrated geospatial microplastic transport
models to assess tire and road wear particles in the Seine watershed. Science of the Total Environment
2019; 646: 1639-1649.
Vaccari DA. Phosphorus: a looming crisis. Scientific American 2009; 300: 54-59.
Van Oost K, Govers G, De Alba S, Quine T. Tillage erosion: a review of controlling factors and implications for
soil quality. Progress in Physical Geography 2006; 30: 443-466.
Van Oost K, Govers G, Desmet P. Evaluating the effects of changes in landscape structure on soil erosion by water
and tillage. Landscape ecology 2000; 15: 577-589.
Van Oost K, Govers G, Quine TA, Heckrath G, Olesen JE, De Gryze S, et al. Landscape-scale modeling of carbon
cycling under the impact of soil redistribution: The role of tillage erosion. Global Biogeochemical Cycles
2005a; 19.
Van Oost K, Quine T, Govers G, Heckrath G. Modeling soil erosion induced carbon fluxes between soil and
atmosphere on agricultural land using SPEROS-C. In: Roose EJ, Lal R, Feller C, Barthes B, Stewart BA,
editors. Advances in soil science. Soil erosion and carbon dynamics. CRC Press, Boca Raton, 2005b, pp.
37-51.
Van Rompaey AJ, Verstraeten G, Van Oost K, Govers G, Poesen J. Modelling mean annual sediment yield using
a distributed approach. Earth Surface Processes and Landforms 2001; 26: 1221-1236.
Verstraeten G, Prosser IP. Modelling the impact of land-use change and farm dam construction on hillslope
sediment delivery to rivers at the regional scale. Geomorphology 2008; 98: 199-212.
Viaroli S, Lancia M, Re V. Microplastics contamination of groundwater: Current evidence and future perspectives.
A review. Science of The Total Environment 2022: 153851.
Wagner S, Hüffer T, Klöckner P, Wehrhahn M, Hofmann T, Reemtsma T. Tire wear particles in the aquatic
environment-a review on generation, analysis, occurrence, fate and effects. Water research 2018; 139: 83-
100.
Waldschläger K, Schüttrumpf H. Infiltration Behavior of Microplastic Particles with Different Densities, Sizes,
and Shapes—From Glass Spheres to Natural Sediments. Environmental Science & Technology 2020; 54:
9366-9373.
Weber CJ, Santowski A, Chifflard P. Investigating the dispersal of macro-and microplastics on agricultural fields
30 years after sewage sludge application. Scientific reports 2022; 12: 1-13.
Weithmann N, Möller JN, Löder MG, Piehl S, Laforsch C, Freitag R. Organic fertilizer as a vehicle for the entry
of microplastic into the environment. Science Advances 2018; 4: eaap8060.
Werkenthin M, Kluge B, Wessolek G. Metals in European roadside soils and soil solution–A review.
Environmental Pollution 2014; 189: 98-110.
Wheeler G, Rolfe G. The relationship between daily traffic volume and the distribution of lead in roadside soil and
vegetation. Environmental Pollution (1970) 1979; 18: 265-274.
Wik A, Dave G. Occurrence and effects of tire wear particles in the environment–A critical review and an initial
risk assessment. Environmental pollution 2009; 157: 1-11.
Witzig C, Wörle K, Földi C, Rehm R, Reuwer A-K, Ellerbrake K, et al. Mikroplastik in Binnengewässern.
Untersuchung und Modellierung des Eintrags und Verbleibs im Donaugebiet als Grundlage für
Maßnahmenplanung. MICBIN Abschlussbericht. 2021.
WRB IWG. World reference base for soil resources 2014, update 2015. Internation soil classification sstem for
naming soils and creating legends for soil maps. World Soil Resources Reports No. 106. FAO 2015.
Wright S, Ulke J, Font A, Chan K, Kelly F. Atmospheric microplastic deposition in an urban environment and an
evaluation of transport. Environment International 2019: 105411.
Zhang L, Xie Y, Liu J, Zhong S, Qian Y, Gao P. An overlooked entry pathway of microplastics into agricultural
soils from application of sludge-based fertilizers. Environmental science & technology 2020; 54: 4248-
4255.



Zhang Y, Gao T, Kang S, Shi H, Mai L, Allen D, et al. Current status and future perspectives of microplastic
pollution in typical cryospheric regions. Earth-Science Reviews 2022; 226: 103924.
Zhao S, Zhang Z, Chen L, Cui Q, Cui Y, Song D, et al. Review on migration, transformation and ecological impacts
of microplastics in soil. Applied Soil Ecology 2022; 176: 104486.
Zhou Y, Wang J, Zou M, Jia Z, Zhou S. Microplastics in soils: A review of methods, occurrence, fate, transport,
ecological and environmental risks. Science of The Total Environment 2020: 141368.
Zubris KA, Richards BK. Synthetic fibers as an indicator of land application of sludge. Environ Pollut 2005; 138:
201-11.