# Peer review of "Model-based analysis of erosion-induced microplastic delivery from arable land to the"

_EGUsphere, 2023_

## Author Response (AR1)

**Reviewer #1:**

Distinguished authors,

thank you for this interesting submission on modelling MP inputs from arable lands to surface waters. The topic is of high relevance for both communities, scientists as well es colleagues involved in practical water resource management.

*Thank you for recognizing the importance of our research topic, and we are grateful for your valuable feedback.*

In your manuscript you are combining a well-established methodology for modelling long-term erosion and sediment transport with MP emissions from various sources, which have the potential to pollute surface waters. The regional study aims at the mesoscale Glonn catchment in Bavaria. As most input data for MP emissions are of coarse resolution or originating from literature and due to the impossible validation, the results of your study show theoretical inputs into the soil and from the upper soil layer into surface waters. Therefore, the value of your study is less in the absolute emissions as much as of showing a potential magnitude and demonstrating systems behaviour for the pathway "water erosion", which is underpinned by scenario analysis.

I have a couple of questions and remarks.

*Thanks for provision!*

What is missing right from the beginning is a definition of MP: MP consists out of a wide range of different polymers with highly differing physical and chemical properties. Which polymers and which of their particle sizes are considered in your study?

*Thank you very much for pointing out that we missed stating right from the beginning which kind of microplastic we address. We had some information later in the model description, but we will undoubtedly give this earlier in the text. In principle, the model does not distinguish between different plastic sizes, types, shapes, etc. but transports everything added to the soil. We could not address more specific microplastic properties based on the rough estimate of MP inputs to soils and several MP sources.*

*In the introduction, a brief general definition of microplastics has now been provided:*

**Microplastic is mostly referred as plastic particles or fibres in a size range of 1 to 5000 μm, originating from the breakdown of larger plastic items or manufacturing at this scale for various purposes (Frias and Nash, 2019; Kim et al., 2021).**

*We have incorporated following explanation into the model description (chapter 2.2) in the beginning of the methods section to explain how MP is considered in our study:*

**The model component responsible for C turnover was not used and focuses exclusively on the erosion, transport, and deposition of C as MP, taking into account the spatially differently distributed MP inputs from different MP sources. As a result of these changes, the model is referred to as SPEROS-MP for the purposes of this study. Based on the model structure it cannot account for particle size-specific selective erosion, and hence, the model does not consider preferential erosion of plastic particles, depending on the size, shape, density, etc. of different polymers. However, the model can account for different transport pathways of different MP input sources e.g., routing tyre wear from fields along streets throughout to catchment to the stream network.**

 "Arable soils in particular experience increased MP inputs as a result of agricultural management (Brandes, 2020)."

"increased" compared to what? Urban areas?

*The sentence refers to the general loading of soils and the general sources in the previous sentence. Compared to non-agricultural soils, agricultural soils experience targeted MP inputs from agricultural management in addition to general MP sources. As a result, agricultural soils have a higher MP input potential. We made it more clearly:*

**Arable soils in particular often experience additional MP inputs associated to agricultural soil management (Brandes, 2020).**

Lines 100-109: Information is missing on the average erosion risk. You are stating the average degree of slope and maximum erosion rates of higher than 10 t/(ha*a), but the understanding of the Glonn catchment would benefit from a map of erosion rates / USLE results. Could you please provide such map?

**A map is provided in Fig 1.** *Average erosion risk is also added:*

**Due to the rolling topography and the erosion-prone soils, a potential long-term mean soil erosion of 5.9 t ha$^{-1}$ a$^{-1}$ (based on the German version of the Universal Soil Loss Equation ABAG) could be calculated for arable land within the catchment (LfL, 2023) with potential erosion rates up to 10 t ha$^{-1}$ a$^{-1}$ (Fig. 1).**

Line 130: The ktc parameter is crucial for determining the sediment delivery and concurrently the MP export pathway. How has it been calculated / derived? In line 159 you state a value of 150 m for ktc: Why do you think is this an appropriate setting for the Glonn catchment? Which measurements in the Glonn catchment did you use to derive the 150 m?

*Ktc values for different land use types must be determined through calibration. The transport capacity coefficient in SPEROS-MP is taken from Van Oost et al. (2003). Based on data from the Belgium Loess Belt, it was calibrated for a 5 m x 5 m grid resolution. We stated it more clear in chapter 2.3.1:*

**The values of the transport capacity coefficient ktc for different land use types must be generally determined through calibration or taken from calibrated literature values (Dlugoß et al., 2012). Based on an extensive study of Van Oost et al. (2003), who tested the sensitivity of the transport capacity coefficient for different arable land and different raster resolutions, an optimum value in case of a 5 m x 5 m grid resolution of ktc = 150 m was determined, which is used in this study. The author (Van Rompaey et al., 2001) identified favorable ktc values ranging between 0 and 60 for non-erosive landscapes at a 20x20 m grid, with an optimum at 42. Given my use of a finer 5x5 m grid resolution, scaled down by a factor of 4, a ktc value of 10 was estimated for forest and grassland.**

*Van Oost, K., Govers, G., & Van Muysen, W. (2003). A process-based conversion model for caesium-137 derived erosion rates on agricultural land: An integrated spatial approach. Earth Surface Processes and Landforms: The Journal of the British Geomorphological Research Group, 28(2), 187-207.*

Line 162: "mean annual precipitation": What is your model period? Are all input data referring to the same period? If not, why?

*We only varied the annual rainfall erosivity as we did not have suitable other input data back to 1950. This is one reason why the model underestimates the annual sediment delivery dynamics. However, as we intended to use the model to show the potential magnitude of MP delivery and also use the model*

*to study the system behaviour, we did not focus more on the input variables (since, by far, the most significant uncertainty in simulated MP delivery results from the estimate of soil contamination based on large-scale input data estimates). The following was added to the model parameters (chapter 2.3.1) for clarification:*

**… Therefore, we followed the approach of Schwertmann et al. (1987) using a relation between annual rainfall erosivity and mean annual precipitation. Based on annual precipitation available in a 1 km x 1 km grid resolution from the German Weather Service (DWD, 2020), yearly R factor maps were created as model input. It is therefore important to note that the variation in model sediment fluxes is solely a result of varying the annual rainfall erosivity, while changes in land management (affecting the C factor of the USLE) are not considered. However, the primary focus of the study was to showcase the potential magnitude and variation of MP delivery, also affected by varying MP input in space and time since 1950. …**

Line 167: The title of your manuscript deals with delivery from arable land, why do you aim at erosion modeling in forest, grassland and settlements?

*We investigate how much MP is delivered from arable soils into the stream network. On the way from the field to the water body, however, we also have to consider the transport through other land uses like forest and grassland. To better evaluate the entry of microplastics into water bodies, we also need to estimate the input in other land uses compared to water, allowing for a more accurate assessment of their relative contributions to water pollution. We also added the information in the text where we are giving the information of the C-factor of the different land uses (chapter 2.3.1):*

*… resulting in a C factor of 0.15, which is constantly used for all arable land in the catchment (Tab. 1).* **Routing sediments from arable land to the stream network, requires a sediment transport through other land uses, like forest, grassland, or paved surfaces. Therefore, these land uses need to be part of the erosion modelling and hence also require a C factor. For forest and grassland, a low C factor of 0.004 and for paved surfaces a C factor of 0.001 was applied (Brandhuber et al., 2018).**

Line 172: ktil is mentioned for the first time but with no background information. Where does this coefficient come from? How is it used in your model? Could you provide a relevant formula? How has it been determined? Why is the value of ktil transferable to the Glonn catchment?

*The tillage erosion coefficient is used to estimate tillage erosion rates. The coefficient represents a literature mean for conventional tillage typically applied in the catchment. The tillage transport coefficient ktil depends on the tillage implement, tillage speed, tillage depths, bulk density, texture, and soil moisture at time of tillage (Van Oost, Govers, et al., 2006). For our study, we used a constant ktil value of 350 kg m$^{-1}$ yr$^{-1}$, as given in Wilken, Ketterer, Koszinski, Sommer, & Fiener (2020). More details in the revised version of the paper are given:*

*…The tillage erosion module of SPEROS-MP follows a diffusion-type equation adopted from Govers et al. (1994) that derives tillage erosion based on change in topography and management-specific coefficients:*

$$Q_{til} = -k_{til} \frac{\Delta h}{\Delta x} \qquad\qquad (Eq.\ 2)$$

*where $Q_{til}$ is the soil flux in kg m$^{-2}$ yr$^{-1}$, $\Delta h$ is the elevation difference in meters, $\Delta x$ is the horizontal distance in meters, and $k_{til}$ is the tillage transport coefficient in kgm$^{-1}$ yr$^{-1}$:*

$$k_{til} = BD_i \cdot TD_i \cdot x_{til} \qquad\qquad (Eq.\ 3)$$

*where $x_{til}$ is the tillage translocation distance in meters, $BD_i$ is the soil bulk density in kg m$^{-3}$, $TD_i$ is the vertical depth of tillage depth (20 cm). It is important to note that…*

Lines 205 ff: Averaging the sewage sludge amounts from the reports over all Bavarian fields including those in the Glonn catchment bears large uncertainties and is practically not a valid method.

The "Klärschlammverordnung" (AbfKlärV) is a very restrictive instrument to manage the transport and distribution of sewage sludge in Germany. According to §6 (1) AbfKlärV from 1992, which is relevant for the time frame you are looking at, arable land can receive up to 5 tonnes per hectare sewage sludge in dried form ("TM"). In practice, only a few parcels receive sewage sludge and most of them don't take sludge. When the sludge is being applied, the parcels receive the full load. After three years a minority receive sludge again, but mainly other parcels are being used then. All this is over stamped by the results from prescribed soil analyses to allow sludge application for on these target parcels.

Therefore, the distribution of sewage sludge is spatially highly variable, and your averaging approach does not reflect the real situation. When you combine this sludge (and MP) average with your erosion rates and sediment deliveries, which are also highly variable in space, then the outcome on mixing in the soil, delivery to streams etc. is very theoretical. A validation of the model results could have revealed this, but it is not possible due to the lack of long-term MP measurements in surface waters and from point and other sources. So, there is no evidence for the validity of your model results.

*You have accurately identified the situation for sewage sludge. However, it must be added that the AbfKlärV has only been available digitally for each parcel since 2011. In the system, the data are without gaps only from 2015. So, we could not make an estimation for sewage sludge for the catchment area back to 1992 or more on a parcel base. However, we do not have parcel-specific information since the 1950s (assuming all associated MP inputs would be more or less stable over time). However, as stated above and in the paper, the intention of the modelling exercise was not to exactly reproduce the MP delivery in the Glonn catchment but to use the model in combination with available input data to perform a system analysis.*

*We tried to make this clearer in der revised version of the manuscript adding more information regarding the sewage sludge input (chapter 2.3.3):*

**Due to the lack of parcel-specific information before 2015 for sewage sludge, we estimated MP inputs using average values per field, similar to compost. However, primary aim in this modelling exercise wasn't to precisely replicate MP delivery in the Glonn catchment. Instead, to demonstrate the model's use in a system analysis, acknowledging limitations in historical data availability.**

Table 2 should be shifted to the end of section 2.3.5 as it summarizes the inputs from all sources considered.

*The table was shifted to the end of section 2.3.5 – thanks for the hint.*

Lines 270-274: The MP input from atmospheric deposition is coupled to the general plastic production in Germany. I don't understand this approach as it bears another source of huge uncertainty.

MP consists out of a wide range of different polymers with highly differing physical and chemical properties. I miss a justification, why the range of MP in your deposition measurements should be equal to the range of plastic polymers being produced in Germany. How are you dealing with this uncertainty?

*The estimation of microplastic (MP) load, based on limited measured data, remains highly uncertain, representing the current data scenario for our analysis. To calculate MP load historically, we relied on*

*the assumption that increased plastic production correlates with higher emissions, although this method is a strong simplification.*

*We acknowledge the need for a more explicit explanation of our MP input estimates and intend to address associated uncertainties in greater detail in our updated manuscript. For instance, our estimate of atmospheric MP deposition since the 1950s using current data and long-term plastic production carries substantial uncertainty. However, our modelling analysis indicates the minor significance of atmospheric input, a point we elaborated on in our revised manuscript:*

*Chapter 2.3.4:*

*For the atmospheric deposition of MP, the data from four bulk deposition measurements (precipitation and dust deposition) in Bavaria (Witzig et al., 2021) were combined with the development of plastics production in Germany since the 1950s.* **Historically, the calculation of MP load relied on the assumption that increased plastic production corresponds to higher emissions (Fig. 2a), although this approach is notably simplified. …**

*Chapter 4.2.1:*

**… In general, taking the considerable uncertainty in the data on MP inputs via the atmosphere into account, the results show that this magnitude is negligible compared to other sources investigated. This finding is important in a scientific context as it provides a better understanding of the magnitude of these inputs. The modelling analysis clearly shows that in comparison to other MP sources the atmospheric inputs are of minimal importance.**

Line 288: "No emissions from unpaved roads and agricultural machinery were considered."

Why not? I think, this would be of importance as it represents a direct input to arable lands.

*We thought about it, but there are no estimates for tire wear on dirt roads and arable soils. In addition, compared with a federal highway, a field is only rarely used and at very low speeds. With soft soil as road surface, we don't think this abrasion is high. However, we know that we are underestimating the MP load with this neglect. We focused on the available data from main traffic.*

Lines 348-351: Please explain in greater detail, why you think that "Given the ongoing increase in plastics production…this may even be a conservative estimate of a business-as-usual scenario pathway."

Most of your MP input comes from tyre wear of adjacent roads. Tyre production is not the same as plastic production and the tyre wear is dependent on traffic density, population numbers and the situation of buffer strips and distances.

*There seems to be a misunderstanding. In the business-as-usual scenario, future projections are based on a fixed value from 2020 (as already stated) without considering the current or future plastic production in Germany. The note that future plastic production has no relation to tire manufacturing is accurate and has been added:*

*Given the ongoing increase in plastics production (Chia et al., 2021; Lwanga et al., 2022)* **and rising traffic numbers (StMB, 2023)**, *this may even be a conservative estimate of a business-as-usual scenario pathway.*

Line 374: You state correctly that the estimated MP inputs contribute significantly to model uncertainty.

How are you dealing with these uncertainties in the model approach and application? Shouldn't the model structure be adapted to these uncertainties?

What are the consequences for the reliability of study results and the usability for practical water resources management?

The uncertainties of the MP input are considered while modelling with different input quantities. For all sources we have presented a minimum, maximum and the mean quantities. The scenarios were also used to investigate the sensitivity of spatial distribution (100-meter distance from the water body).

For the water management, we can also only provide estimates. But the aim of the study was to better assess the role of soil erosion in the MP cycle. Until now, it was unclear what the extent of soil erosion could be for MP input to water bodies. The fact that in rural areas more MP could enter water bodies through soil erosion than, for example, through wastewater treatment plant outlets is an important finding. Water managers know from our results that soil erosion is not negligible with respect to MP.

Figure 6: Inserting the road network more clearly would help to understand the spatial distribution of polluted areas. Maybe this is a visibility problem with the resolution of the graphics.

*Thanks for the hint – the road network was added.*

Please explain, why you cannot see in the map the higher MP load in those parcels close to roads, which receive the high input from the tyre wear in comparison to parcels farer away.

*The difference is shown in the figure units b) and c). While in b) (MP input sewage sludge, compost and atmospheric input) the MP input is evenly distributed on all arable land, in c) (tire wear) the areas are loaded differently and become less MP with the distance to the road. Some croplands get nothing at all. With the now visible roads, it should be recognizable.*

Line 493: Sediment transport modelling is of course a difficult topic. But you should rethink, if you evaluate a R² of 51 % as "perfectly"? In line 366 you are talking about "satisfactorily".

What is true now: perfection or satisfaction?

*You are right. Even if the long-term average is "perfectly" reproduced, we should be more careful with the wording considering an $R^2$ of 51%. Despite the range of the individual years, the long-term average is well represented. We do not use "perfectly" anymore.*

A formal point: Years in the references are lacking a closing bracket quite often throughout the manuscript. Please correct.

*Thank you, we checked it.*

I would suggest to implement two additional aspects, maybe in the discussion:

1. Arable land is contributing to MP pollution in surface waters, BUT the major inputs into arable soils are coming from non-agricultural sources. What can be done to reduce inputs from tyre wear?

*This has been correctly identified. Measures to prevent MP in soil will have little noticeable effect if TW remains unchanged. We added a paragraph to the discussion in chapter 4.2.2:*

***It should be noted that TW as not-agriculture MP-source is of paramount importance compared to other MP sources, especially with respect to MP reduction measures. Not only for soil, but also for water bodies and probably all other environmental compartments. Measures to prevent MP in soil will have little noticeable effect if TW remains unchanged. This problem should be given more consideration in future studies and interpretation of results (Knight et al., 2020a; Knight et al., 2020b; Mennekes and Nowack, 2022).***

2. Plastic pollution in all environmental compartments is a major challenge. By far more measurements and basic research in this field are required to foster process understanding and to improve model applications significantly.

*We fully agree with this comment. It is challenging and much more research is necessary reaching from improvements of measurement techniques (incl. standardisation, faster measurements etc.) to field studies regarding the behaviour of MP in the soil system. We gave some hints regarding the challenges and knowledge gaps the discussion (chapter 4.3):*

***This analysis not only sheds light on the model's impact on MP distribution in varied landscape contexts but also underscores the potential environmental repercussions. The study significantly advances scientific understanding and practical relevance by emphasizing long-term field experiments and meso-scale model analyses. Nevertheless, gaps persist in MP research, particularly concerning standardized detection methods and quantification of terrestrial MP pollution. Addressing these gaps requires extensive additional research to comprehensively grasp the scope of MP pollution across diverse environmental media and the entirety of the MP cycle. Substantial measurements and fundamental research in this domain are imperative to enhance process comprehension and refine model applications.***

We thank you for your kind and constructive feedback on our manuscript, which we greatly appreciate.
* * *
**Reviewer #2:**

This study addresses an interesting question by evaluating the contribution of erosion processes to microplastic delivery to the river network. I applaud the authors to collate a substantial amount of information on different sources of plastic inputs into soils. Overall, the paper is well written, the results are clearly described, and an in-depth analysis of the results is provided.

*We thank the reviewer for this positive feedback and appreciate his/her efforts in taking the time provide constructive comments.*

However, I do feel that the methods are not fully appropriate: the use of a complex spatially- and temporally explicit model in a context where no model-constraining data is available is questionable. All plastic pathways are proportional to sediment fluxes and can therefore be inferred directly from sediment pathways. There are very large uncertainties associated with the estimation of the plastic input (info on spatial and temporal patterns is absent) and several key processes are not constrained

or represented in the model (e.g. size selectivity of detachment, transport and deposition, plastic weathering, ….). Together, this makes its application to a specific case-study in a low-data context challenging.

*We agree with the reviewer that data availability on plastic inputs to agricultural soils is challenging. We have therefore invested considerable time and energy in developing reasonable estimates of plastic inputs from various sources. This includes estimates of the associated uncertainties for each input pathway. However, as the input estimates obviously are associated with large uncertainties, we did not include aspects of plastic fragmentation as we assume that most plastic might fragment but will not be degraded in the time spans, we are evaluating. Regarding spatio-temporal patterns of MP input into the soils we included some spatial and temporal variability: (i) For sewage sludge, compost, and atmospheric deposition we assumed homogeneous inputs to all fields as we did not have long-term field-specific input data. Nevertheless, we accounted for a yearly variability in the different input pathways. Moreover, for the scenarios, different assumptions were made regarding the spatial distribution of MP inputs. (ii) For tire wear (the most important MP input pathway) we did account for spatially variable MP input. The tire wear MP input was allocated to the specific fields along individual roads, whereas the traffic load of the roads was individually considered.*

*Additionally, to this explanation* **we have incorporated various clarifications into the study text** *also based on Review #1's suggested changes, addressing the mentioned concerns to enhance clarity.*

I understand that a scoping study can rely on several assumptions and simplifications. However, I suggest that the authors evaluate to what extent their complex approach is justifiable under the absence of sufficient quality input and validation data. An alternative, and likely more robust approach is to use a simple (spatially lumped) mass-balance or accounting model that allows to cover both a range of plastic input- and a range of water, tillage, sediment delivery- scenarios. This could provide a more comprehensive assessment of plastic pathways, residence times and delivery. I am confident that the authors can re-frame this study and maybe add a more robust approach.

*We do not agree that the use of a (complex) spatially and temporally explicit model is questionable because it does provide more insight than a lumped model. We do not agree for the following reasons:*

*(i) As mentioned above, we have assumed spatially and temporally variable MP inputs and considered MP accumulation increasing since 1950. Hence, a spatially and temporally explicit model is essential to account for this spatio-temporal dynamics in MP inputs.*

(ii) As the model is routing sediments and MP through the landscape while including deposition of both it is creating a complex pattern of MP concentration in up to 10 soil layers (10 cm each to a max depths of 1 m), which is updated once a year. In consequence, MP can be for example trapped in grassed areas or buried below the plough layer due to water and tillage erosion. Both is interesting to be analysed, as for example the MP trapping in grassed areas along streams might later on lead to MP leaching to shallow ground water. However, from the reviewer's comments on the model used and its results, we learned that our explanation of the model and its dynamic results was not sufficient, e.g. the model does not produce an MP flux proportional to the sediment flux, as misinterpreted by the reviewer.

*Therefore, we revised our manuscript to clarify why our chosen modelling approach is appropriate and necessary for addressing the specific research questions in our study. To the methods in chapter 2.2 was added:*

*The erosion and MP transport is modelled based on a modified version of the spatially distributed water and tillage erosion and carbon (C) turnover model SPEROS-C (Fiener et al., 2015; Van Oost et al., 2005a).* **SPEROS-C was deliberately selected as (i) it allows the spatially explicit integration of yearly MP inputs since 1950, (ii) it routs sediment and MP through the landscape while including deposition of both, and (iii) it includes water and tillage erosion as well as a yearly soil profile update (10 layers of**

***10 cm thickness) accounting for changes in MP allocation following erosion or deposition. Both, the modeled deposition and the MP soil profile update allow us to analyze potential MP landscape sinks either in space (e.g., in grassed areas) or in depths below the plough layer, where MP is not affected by water erosion anymore.***

*The model was originally developed to analyse the long-term effect of soil erosion on landscape-scale carbon balance…*
* * *
*We sincerely appreciate your feedback, which has helped us reflect on our methodology and its communication in the paper.*

---

## Author Response (AR2)

**Reviewer #1:**

Distinguished authors,

thank you for this interesting submission on modeling MP inputs from arable lands to surface waters. The topic is of high relevance for both communities, scientists as well es colleagues involved in practical water resource management.

*Thank you for recognizing the importance of our research topic, and we are grateful for your valuable feedback.*

In your manuscript you are combining a well established methodology for modeling long-term erosion and sediment transport with MP emissions from various sources, which have the potential to pollute surface waters. The regional study aims at the mesoscale Glonn catchment in Bavaria. As most input data for MP emissions are of coarse resolution or originating from literature and due to the impossible validation, the results of your study show theoretical inputs into the soil and from the upper soil layer into surface waters. Therefore, the value of your study is less in the absolute emissions as much as of showing a potential magnitude and demonstrating systems behaviour for the pathway "water erosion", which is underpinned by scenario analysis.

I have a couple of questions and remarks.

*Thanks for provision!*

What is missing right from the beginning is a definition of MP: MP consists out of a wide range of different polymers with highly differing physical and chemical properties. Which polymers and which of their particle sizes are considered in your study?

*Thank you very much for pointing out that we missed stating right from the beginning which kind of microplastic we address. We had some information later in the model description, but we will undoubtedly give this earlier in the text. In principle, the model does not distinguish between different plastic sizes, types, shapes, etc. but transports everything added to the soil. We could not address more specific microplastic properties based on the rough estimate of MP inputs to soils and several MP sources.*

*In the introduction, a brief general definition of microplastics has now been provided:*

**Speaking of microplastic, we refer to tiny plastic particles, less than 5 mm, that originate from the breakdown of larger plastic items or are manufactured at a small scale for various purposes (Frias and Nash, 2019; Kim et al., 2021).**

*We have incorporated following explanation into the model description (chapter 2.2) in the beginning of the methods section to explain how MP is considered in our study:*

**The SPEROS-C model operates on a mass-balanced principle, calculating C input in mass (kg m²). For the purposes of this study, the C turnover component of SPEROS-C was not utilized, leading to the adaptation of the model as SPEROS-MP. In its MP-specific iteration, SPEROS-MP estimates MP input in mass (kg m²). Consequently, the model does not account for the specific characteristics such as polymer type, particle number, size, shape, density, or chemical properties of MP particles from various sources. It treats the erodibility of MP from all input pathways equally, aligning with its approach of considering all potential microplastics without differentiation based on their properties.**

Lines 34/35: "Arable soils in particular experience increased MP inputs as a result of agricultural management (Brandes, 2020)."

"increased" compared to what? Urban areas?

*The sentence refers to the general loading of soils and the general sources in the previous sentence. Compared to non-agricultural soils, agricultural soils experience targeted MP inputs from agricultural management in addition to general MP sources. As a result, agricultural soils have a higher MP input potential. We made it more clearly:*

**Arable soils in particular often experience increased MP inputs due to agricultural management and associated additional MP sources, compared to soils not used for agricultural purposes. (Brandes, 2020).**

Lines 100-109: Information is missing on the average erosion risk. You are stating the average degree of slope and maximum erosion rates of higher than 10 t/(ha*a), but the understanding of the Glonn catchment would benefit from a map of erosion rates / USLE results. Could you please provide such map?

*A map is provided in **Fig 1**. Average erosion risk is also added:*

**Due to the topography and the soils, an average, long-term soil erosion of 5.9 t ha-1 a-1 (based on the German version of the Universal Soil Loss Equation ABAG) could be calculated for arable land (Auerswald et al., 2009; LfL, 2023) with erosion rates up to 10 t ha-1 a-1 (Fig. 1).**

Line 130: The ktc parameter is crucial for determining the sediment delivery and concurrently the MP export pathway. How has it been calculated / derived? In line 159 you state a value of 150 m for ktc: Why do you think is this an appropriate setting for the Glonn catchment? Which measurements in the Glonn catchment did you use to derive the 150 m?

*Ktc values for different land use types must be determined through calibration. The transport capacity coefficient in SPEROS-MP is taken from Van Oost et al. (2003). Based on data from the Belgium Loess Belt, it was calibrated for a 5 m x 5 m grid resolution. We stated it more clear in chapter 2.3.1:*

**Ktc values for different land use types must be determined through calibration (Dlugoß et al., 2012). A transport capacity coefficient ktc of 150 m was used as the optimum value for cropland for a 5 m x 5 m grid resolution, calibrated by Van Oost et al. (2003).**

*Van Oost, K., Govers, G., & Van Muysen, W. (2003). A process-based conversion model for caesium-137 derived erosion rates on agricultural land: An integrated spatial approach. Earth Surface Processes and Landforms: The Journal of the British Geomorphological Research Group, 28(2), 187-207.*

Line 162: "mean annual precipitation": What is your model period? Are all input data referring to the same period? If not, why?

*We only varied the annual rainfall erosivity as we did not have suitable other input data back to 1950. This is one reason why the model underestimates the annual sediment delivery dynamics. However, as we intended to use the model to show the potential magnitude of MP delivery and also use the model to study the system behavior, we did not focus more on the input variables (since, by far, the most significant uncertainty in simulated MP delivery results from the estimate of soil contamination based on large-scale input data estimates). The following was added to the model parameters (chapter 2.3.1) for clarification:*

**The variation in the model is limited to altering the annual rainfall erosivity due to insufficient available input data dating back to 1950. However, the primary focus of the study was to showcase the**

*potential magnitude of MP delivery and to explore system behavior. The variability in annual precipitation erosivity is determined by the annual average of precipitation*

Line 167: The title of your manuscript deals with delivery from arable land, why do you aim at erosion modeling in forest, grassland and settlements?

*We investigate how much MP is delivered from arable soils into the stream network. On the way from the field to the water body, however, we also have to consider the transport through other land uses like forest and grassland. To better evaluate the entry of microplastics into water bodies, we also need to estimate the input in other land uses compared to water, allowing for a more accurate assessment of their relative contributions to water pollution. We also added the information in the text where we are giving the information of the C-factor of the different land uses (chapter 2.3.1):*

*… resulting in a C factor of 0.15, which is constantly used for all arable land in the catchment (Tab. 1).* **On the way from the field to the water body, however, the transport through other land uses like forest and grassland has to be considered.** *In the case of forest and grassland, a low C factor of 0.004 and for settlements a C factor of 0.001 was applied…*

Line 172: ktil is mentioned for the first time but with no background information. Where does this coefficient come from? How is it used in your model? Could you provide a relevant formula? How has it been determined? Why is the value of ktil transferable to the Glonn catchment?

*The tillage erosion coefficient is used to estimate tillage erosion rates. The coefficient represents a literature mean for conventional tillage typically applied in the catchment. The tillage transport coefficient ktil depends on the tillage implement, tillage speed, tillage depths, bulk density, texture, and soil moisture at time of tillage (Van Oost, Govers, et al., 2006). For our study, we used a constant ktil value of 350 kg m⁻¹ yr⁻¹, which was recently determined for this region (Wilken, Ketterer, Koszinski, Sommer, & Fiener, 2020). More details in the revised version of the paper are given:*

*…The tillage erosion module of SPEROS-Pu follows a diffusion-type equation adopted from Govers et al. (1994) that derives tillage erosion based on change in topography and management-specific coefficients:*

$$Q_{til} = -k_{til}\, \frac{\Delta h}{\Delta x} \qquad\qquad (Eq.\ 2)$$

*where $Q_{til}$ is the soil flux in $kgm^{-2}\ yr^{-1}$, $\Delta h$ is the elevation difference in metres, $\Delta x$ is the horizontal distance in metres, and $k_{til}$ is the tillage transport coefficient in $kgm^{-1}\ yr^{-1}$:*

$$k_{til} = BD_i \cdot TD_i \cdot x_{til} \qquad\qquad (Eq.\ 3)$$

*where $x_{til}$ is the tillage translocation distance in metres, $BD_i$ is the soil bulk density in $kgm^{-3}$, $TD_i$ is the vertical depth of tillage depth (20 cm). For the Glonn catchment, we used a constant ktil value of 350 kg m-1 yr-1, which was determined for another loess dominated region within Germany by Wilken et al. (2020).*

Lines 205 ff: Averaging the sewage sludge amounts from the reports over all Bavarian fields including those in the Glonn catchment bears large uncertainties and is practically not a valid method.

The "Klärschlammverordnung" (AbfKlärV) is a very restrictive instrument to manage the transport and distribution of sewage sludge in Germany. According to §6 (1) AbfKlärV from 1992, which is relevant for the time frame you are looking at, arable land can receive up to 5 tonnes per hectare sewage sludge in dried form ("TM"). In practice, only a few parcels receive sewage sludge and most of them don't take sludge. When the sludge is being applied, the parcels receive the full load. After three years a

minority receive sludge again, but mainly other parcels are being used then. All this is overstamped by the results from prescribed soil analyses to allow sludge application for on these target parcels.

Therefore, the distribution of sewage sludge is spatially highly variable and your averaging approach does not reflect the real situation. When you combine this sludge (and MP) average with your erosion rates and sediment deliveries, which are also highly variable in space, then the outcome on mixing in the soil, delivery to streams etc. is very theoretical. A validation of the model results could have revealed this, but it is not possible due to the lack of long-term MP measurements in surface waters and from point and other sources. So, there is no evidence for the validity of your model results.

*You have accurately identified the situation for sewage sludge. However, it must be added that the AbfKlärV has only been available digitally for each parcel since 2011. In the system, the data are without gaps only from 2015. So, we could not make an estimation for sewage sludge for the catchment area back to 1992 or more on a parcel base. However, we do not have parcel-specific information since the 1950s (assuming all associated MP inputs would be more or less stable over time). However, as stated above and in the paper, the intention of the modeling exercise was not to exactly reproduce the MP delivery in the Glonn catchment but to use the model in combination with available input data to perform system analysis.*

*We made it more evident in der revised version of the manuscript adding these information in the methods for sewage sludge data (chapter 2.3.3):*

***Due to the lack of parcel-specific information before 2015 for sewage sludge, we calculated estimates using average values per field, similar to compost, assuming stability in MP inputs over time. However, primary aim in this modeling exercise wasn't to precisely replicate MP delivery in the Glonn catchment. Instead, to demonstrate the model's use in system analysis, acknowledging limitations in historical data availability.***

Table 2 should be shifted to the end of section 2.3.5 as it summarizes the inputs from all sources considered.

*The table was shifted to the end of section 2.3.5 – thanks for the hint.*

Lines 270-274: The MP input from atmospheric deposition is coupled to the general plastic production in Germany. I don't understand this approach as it bears another source of huge uncertainty.

MP consists out of a wide range of different polymers with highly differing physical and chemical properties. I miss a justification, why the range of MP in your deposition measurements should be equal to the range of plastic polymers being produced in Germany. How are you dealing with this uncertainty?

*The estimation of microplastic (MP) load, based on limited measured data, remains highly uncertain, representing the current data scenario for our analysis. To calculate MP load historically, we relied on the assumption that increased plastic production correlates with higher emissions, although this method is significantly simplified.*

*We acknowledge the need for a more explicit explanation of our MP input estimates and intend to address associated uncertainties in greater detail in our updated manuscript. For instance, our estimate of atmospheric MP deposition since the 1950s using current data and long-term plastic production carries substantial uncertainty. However, our modeling analysis indicates the minor significance of atmospheric input, a point we elaborated on in our revised manuscript:*

*Chapter 2.3.4:*

*For the atmospheric deposition of MP, the data from four bulk deposition measurements (precipitation and dust deposition) in Bavaria (Witzig et al., 2021) were combined with the development of plastics production in Germany since the 1950s. **Historically, the calculation of MP load relied on the assumption that increased plastic production corresponds to higher emissions (Fig. 2a), although this approach is notably simplified …***

*Chapter 4.2.1:*

***… In general, taking the considerable uncertainty in the data on MP inputs via the atmosphere into account, the results show that this magnitude is negligible compared to other sources investigated. This finding is important in a scientific context as it provides a better understanding of the magnitude of these inputs. The modeling analysis emphatically shows that atmospheric inputs are of minimal importance in comparison.***

Line 288: "No emissions from unpaved roads and agricultural machinery were considered."

Why not? I think, this would be of importance as it represents a direct input to arable lands.

*We thought about it, but there are no estimates for tire wear on dirt roads and arable soils. In addition, compared with a federal highway, a field is only rarely used and at very low speeds. With soft soil as road surface, we don't think this abrasion is high. However, we know that we are underestimating the MP load with this neglect. We focused on the available data from main traffic.*

Lines 348-351: Please explain in greater detail, why you think that "Given the ongoing increase in plastics production…this may even be a conservative estimate of a business-as-usual scenario pathway."

Most of your MP input comes from tyre wear of adjacent roads. Tyre production is not the same as plastic production and the tyre wear is dependent on traffic density, population numbers and the situation of buffer strips and distances.

*There seems to be a misunderstanding. In the business-as-usual scenario, future projections are based on a fixed value from 2020 (as already stated) without considering the current or future plastic production in Germany. The note that future plastic production has no relation to tire manufacturing is accurate and has been added:*

*Given the ongoing increase in plastics production (Chia et al., 2021; Lwanga et al., 2022) **and rising traffic numbers (StMB, 2023)**, this may even be a conservative estimate of a business-as-usual scenario pathway.*

Line 374: You state correctly, that the estimated MP inputs contribute significantly to model uncertainty.

How are you dealing with these uncertainties in the model approach and application? Shouldn't the model structure be adapted to these uncertainties?

What are the consequences for the reliability of study results and the usability for practical water resources management?

The uncertainties of the MP input are taken into account by modeling with different input quantities. For all sources we have presented a minimum, maximum and the mean quantities. The scenarios were also used to investigate the sensitivity of spatial distribution (100 meter distance from the water body).

For the water management, we can also only provide estimates. But, the aim of the study was to better assess the role of soil erosion in the MP cycle. Until now, it was unclear what the extent of soil erosion could be for MP input to water bodies. The fact that in rural areas more MP could enter water bodies through soil erosion than, for example, through wastewater treatment plant outlets is an important finding. Water managers know from our results that soil erosion is not negligible with respect to MP. Preventive measures may need to be taken for waters with some order of protection.

Figure 6: Inserting the road network more clearly would help to understand the spatial distribution of polluted areas. Maybe this is a visibility problem with the resolution of the graphics.

*Thanks for the hint – the road network was added.*

Please explain, why you cannot see in the map the higher MP load in those parcels close to roads, which receive the high input from the tyre wear in comparison to parcels farer away.

*The difference is shown in the figure units b) and c). While in b) (MP input sewage sludge, compost and atmospheric intput) the MP input is evenly distributed on all arable land, in c) (tire wear) the areas are loaded differently and become less MP with the distance to the road. Some croplands get nothing at all. With the now visible roads, it should be recognizable.*

Line 493: Sediment transport modeling is of course a difficult topic. But you should rethink, if you evaluate a R² of 51 % as "perfectly"? In line 366 you are talking about "satisfactorily".

What is true now: perfection or satisfaction?

*You are right. Even if the long-term average is "perfectly" reproduced, we should be more careful with the wording considering an R² of 51%. Despite the range of the individual years, the long-term average is well represented. We do not use "perfectly" anymore.*

A formal point: Years in the references are lacking a closing bracket quite often throughout the manuscript. Please correct.

*Thank you, we checked it.*

I would suggest to implement two additional aspects, maybe in the discussion:

1. Arable land is contributing to MP pollution in surface waters, BUT the major inputs into arable soils are coming from non-agricultural sources. What can be done to reduce inputs from tyre wear?

    *This has been correctly identified. Measures to prevent MP in soil will have little noticeable effect if TW remains unchanged. We added a paragraph to the discussion in chapter 4.2.2:*

    ***It should be noted that TW as not-agriculture MP-source is of paramount importance compared to other MP sources, especially with respect to MP reduction measures. Not only for soil, but also for water bodies and probably all other environmental compartments. Measures to prevent MP in soil will have little noticeable effect if TW remains unchanged.***

*This problem should be given more consideration in future studies and interpretation of results (Knight et al., 2020a; Knight et al., 2020b; Mennekes and Nowack, 2022).*

2. Plastic pollution in all environmental compartments is a major challenge. By far more measurements and basic research in this field are required to foster process understanding and to improve model applications significantly.

   *The discussion is specifically focused on the issue of MP and arable soils. This is the subject of the paper. We want to avoid stretching the discussion with general arguments. But the hint is directional and we decided to take it up in the Conclusion:*

   ***Overall, the work contributes significantly to scientific knowledge and practical relevance through its focus on long-term studies, field experiments under real conditions, and a meso-scale model analysis. However, there are still deficits in MP research, especially in terms of standardized detection methods and quantification of MP pollution in the terrestrial environment. There is still a significant need for further research to better under-stand the full extent of MP pollution in various environmental media and the entire MP cycle. By far more measurements and basic research in this field are required to foster process understanding and to improve model applications significantly.***

We thank you for your kind and constructive feedback on our manuscript, which we greatly appreciate.
* * *
**Reviewer #2:**

This study addresses an interesting question by evaluating the contribution of erosion processes to microplastic delivery to the river network. I applaud the authors to collate a substantial amount of information on different sources of plastic inputs into soils. Overall, the paper is well written, the results are clearly described, and an in-depth analysis of the results is provided.

*We thank the reviewer for this positive feedback and appreciate his/her efforts in taking the time provide constructive comments.*

However, I do feel that the methods are not fully appropriate: the use of a complex spatially- and temporally explicit model in a context where no model-constraining data is available is questionable. All plastic pathways are proportional to sediment fluxes and can therefore be inferred directly from sediment pathways. There are very large uncertainties associated with the estimation of the plastic input (info on spatial and temporal patterns is absent) and several key processes are not constrained or represented in the model (eg size selectivity of detachment, transport and deposition, plastic weathering, ….). Together, this makes its application to a specific case-study in a low-data context challenging.

*We agree with the reviewer that data availability on plastic inputs to agricultural soils is challenging. We have therefore invested considerable time and energy in developing reasonable estimates of plastic*

*inputs from various sources. This includes estimates of the associated uncertainties for each input path-way. However, as the input estimates obviously are associated with large uncertainties, we did not include aspects of plastic fragmentation as we assume that most plastic might fragment but will not be degraded in the time spans we are evaluating. Regarding spatio-temporal patterns of MP input into the soils we included some spatial and temporal variability: (i) For sewage sludge, compost, and atmospheric deposition we assumed homogeneous inputs to all fields as we did not have field-specific input data. Nevertheless, we accounted for a yearly variability in the different input pathways. Moreover, for the scenarios, different assumptions were made regarding the spatial distribution of MP inputs. (ii) For tire wear (the most important MP input pathway) we did account for spatially variable MP input. The tire wear MP input was allocated to the specific fields along individual roads, whereas the traffic load of the roads was individually considered.*

*Additionally, to this explanation **we have incorporated various clarifications into the study text** also based on Review #1's suggested changes, addressing the mentioned concerns to enhance clarity.*

I understand that a scoping study can rely on several assumptions and simplifications. However, I suggest that the authors evaluate to what extent their complex approach is justifiable under the absence of sufficient quality input and validation data. An alternative, and likely more robust approach is to use a simple (spatially lumped) mass-balance or accounting model that allows to cover both a range of plastic input- and a range of water, tillage, sediment delivery- scenarios. This could provide a more comprehensive assessment of plastic pathways, residence times and delivery. I am confident that the authors can re-frame this study and maybe add a more robust approach.

*We do not agree that the use of a (complex) spatially and temporally explicit model is questionable because it does not provide more insight than a lumped model. We do not agree for the following reasons:*

*(i) As mentioned above, we have assumed spatially and temporally variable MP inputs and considered MP accumulation increasing since 1950. Hence, a spatially and temporally explicit model is essential to account for this spatio-temporal dynamics in MP inputs.*

(ii) As the model is routing sediments and MP through the landscape while including deposition of both it is creating a complex pattern of MP concentration in up to 10 soil layers (10 cm each to a max depths of 1 m), which is updated once a year. In consequence, MP can be for example trapped in grassed areas or buried below the plough layer due to water and tillage erosion. Both is interesting to be analyzed, as for example the MP trapping in grassed areas along streams might lead to MP leaching to shallow ground water. However, from the reviewer's comments on the model used and its results, we learned that our explanation of the model and its dynamic results was not sufficient, e.g. the model does not produce an MP flux proportional to the sediment flux, as misinterpreted by the reviewer.

*Therefore, we revised our manuscript to clarify why our chosen modeling approach is appropriate and necessary for addressing the specific research questions in our study. To the methods in chapter 2.2 was added:*

*The erosion and MP transport is modelled based on a modified version of the spatially distributed water and tillage erosion and carbon (C) turnover model SPEROS-C (Fiener et al., 2015; Van Oost et al., 2005a). **SPEROS-C was deliberately selected as a spatially and temporally explicit model for specific reasons. The use of a spatially and temporally explicit model is essential due to our assumption of spatially and temporally variable MP inputs and the consideration of MP accumulation since 1950, enabling a comprehensive integration of these spatio-temporal dynamics into the analysis. The model's ability to transport sediments and microplastics across the landscape, including deposition mechanisms, generates detailed, annually updated concentration patterns of microplastics in soil layers from 10 cm thick to one meter deep, illustrating potential scenarios for microplastic deposition to other land***

*uses such as grasslands or burial under plow layers through erosion processes. These scenarios illustrate potential scenarios for microplastic deposition in other land uses such as grasslands or burial under the plow layer by erosion processes and allow a comprehensive analysis of microplastic behavior, including potential leaching into shallow groundwater from grasslands near streams.The model was originally developed to analyse the long-term effect of soil erosion on landscape-scale carbon balance…*

*We sincerely appreciate your feedback, which has helped us reflect on our methodology and its communication in the paper.*
* * *
**Corrections**

Thank you for the detailed and adequate revisions you made following the reviewers' comments. These helped to clarify and strengthen the message of the paper. I have a few remaining editorial comments.

Additional private note (visible to authors and reviewers only):
I have a few comments regarding units. If I understood it correctly, you found that burying op MPs at locations where sedimation occurred was important for the immobilisation of MPs. But, this burying is not a transport process of these particles into the subsoil. In order to avoid confusion, I propose not to use 'transport' here. Another term that generated some confusion is 'conservative'. This could have opposite meanings. In a risk assessment perspective, conservative means that your estimate should likely to be an overestimation of the MP load to surface water. But, I assume you meant it oppositely. Therefore I propose to articulate clearly what is meant by 'conservative' or avoid using it. Below you find detailed editorial comments.

*Thank you for the very constructive feedback! The suggestions have been accepted and implemented to enhance the clarity of the paper's statements*

Ln 148: check units. Shouldn't ktc have unit m?

*Unit m was added*

Ln 156: check the units of Qtil and ktil. They should be identical.

*Unit of Qtil was corrected*

Ln 160: Add units of TD.

*Unit m was added*

Ln 207-208: add units to the ktc factors you mention.

*Unit m was added*

Ln 221: what is an 'exclusive input estimate'? Do you mean, input was exclusively estimated for arable land?

*The sentence was clarified: "**As soil erosion is dominant on arable land, an MP input estimate was solely performed for arable land."***

Ln 227: '… did separate and combined modelling runs for the different contamination estimates' This is quite unclear at this point in the text. What are 'separate' and 'combined' modelling runs? What is separated or combined?

*Additional information was added: did separate (for each source) and combined (for the sum of all sources) modelling runs*

Ln 272: Fig 2 a) check y axis label: should be 'emission'

*Y axis lable was corrected*

Ln 445: 'For comparison, the amount of MP delivery through wastewater treatment plants (WWTP) in 2020.' This was mentioned already before and can be skipped here.

*The duplication was removed*

Ln 467: I thought you did not simulate translocation of MPs with soil water flow. Therefore, I think it is better to write that MPs were 'buried' below the plough layer rather than that they were 'moved' below the plough layers. This only occurs when deposition of sediments leads to increase of the surface elevation.

*Buried was written*

Ln 470: Here also, replace 'translocated' by 'buried'.

*Buried was written*

Ln 471: 'This transport into the subsoil was caused by water erosion (48.5%) and tillage erosion (51.5%).' I do not really understand this. Isn't this again a burial rather than a transport? Second, isn't it rather related to a sedimentation than erosion? The sediment originates from erosion.

*The sentece was specified: This burial into the subsoil was caused by sedimentation via water erosion (48.5%) and tillage erosion (51.5%)*

Ln 473: change infiltration to leaching. Infiltration would correspond to water flowing into the soil layers and hence input of MPs rather than export of MPs.

*Leaching was taken*

Ln 529: Due you mean that you overestimate the number of years with a low delivery and underestimate the number of years with a high delivery? Or do you overestimate the delivery in years with a low delivery and vice versa for years with a high delivery?

*We specified in the chapter and added: "Averaging the model input variables led to an overestimation of years with low sediment delivery and an underestimation of years with high sediment delivery (Fig. 4)."*

Ln 584: add 'via erosion from arable land'. I suppose that an important fraction would also arrive in the river from wash off from the roads and the grass strips along the roads.

*The addition was inserted*

Ln 599: It is not clear to me what you mean by 'conservative' here. In a risk assessment, you would call the estimate 'conservative' when it would be in the higher range of inputs so that the estimated

input is unlikely to be higher in reality. But is this what you would mean here? I think you should make clearer whether you mean that the input estimate might be too low or too high.

*The statement was specified: In comparison, when focusing on the erosion of MPs mixed into the plough layer, only 0.11% of the applied TW to arable soils from 1950 to 2020 reached the river system. While TW represents the largest entry source in our study, the overall MP flow to the stream network is an underestimation given the simplified approach.*

Ln 680: In view of risk assessment, the estimate would in fact not be conservative since there is a possibility that it is too low.

*Sentence was clarified: In this regard, the modeled MP delivery is therefore an underestimated estimation.*

Ln 697: since erosion 'and tillage' incorporate MP-free subsoil in the tilled surface layer.

*Was added*